# Collaborating with Humans without Human Data

**DJ Strouse**,[*] **Kevin R. McKee, Matt Botvinick, Edward Hughes, Richard Everett**[*]
DeepMind
{strouse, kevinrmckee, botvinick, edwardhughes, reverett}@deepmind.com

## Abstract

Collaborating with humans requires rapidly adapting to their individual strengths, weaknesses, and preferences. Unfortunately, most standard multi-agent reinforcement learning techniques, such as self-play (SP) or population play (PP), produce agents that overfit to their training partners and do not generalize well to humans. Alternatively, researchers can collect human data, train a human model using behavioral cloning, and then use that model to train "human-aware" agents ("behavioral cloning play", or BCP). While such an approach can improve the generalization of agents to new human co-players, it involves the onerous and expensive step of collecting large amounts of human data first. Here, we study the problem of how to train agents that collaborate well with human partners without using human data. We argue that the crux of the problem is to produce a diverse set of training partners. Drawing inspiration from successful multi-agent approaches in competitive domains, we find that a surprisingly simple approach is highly effective. We train our agent partner as the best response to a population of self-play agents and their past checkpoints taken throughout training, a method we call Fictitious Co-Play (FCP). Our experiments focus on a two-player collaborative cooking simulator that has recently been proposed as a challenge problem for coordination with humans. We find that FCP agents score significantly higher than SP, PP, and BCP when paired with novel agent and human partners. Furthermore, humans also report a strong subjective preference to partnering with FCP agents over all baselines.

## 1 Introduction

Generating agents which collaborate with novel partners is a longstanding challenge for Artificial Intelligence (AI) [4, 16, 37, 52]. Achieving ad-hoc, zero-shot coordination [31, 66] is especially important in situations where an AI must generalize to novel human partners [6, 61]. Many successful approaches have employed human models, either constructed explicitly [14, 35, 53] or learnt implicitly [12, 60]. By contrast, recent work in competitive domains has shown that it is possible to reach human-level using model-free reinforcement learning (RL) without human data, via self-play [8, 9, 63, 64]. This begs the question: Can model-free RL without human data generate agents that can collaborate with novel humans?

We seek an answer to this question in the space of common-payoff games, where all agents work towards a shared goal and receive the same reward. Self-play (SP), in which an agent learns from repeated games played against copies of itself, does not produce agents that generalize well to novel co-players [10, 11, 21, 44]. Intuitively, this is because agents trained in self-play only ever need to coordinate with themselves, and so make for brittle and stubborn collaborators with new partners who act differently. Population play (PP) trains a population of agents, all of whom interact with each other [39]. While PP can generate agents capable of cooperation with humans in competitive team games [34], it still fails to produce robust partners for novel humans in pure common-payoff settings

---

[*]Equal contribution.

35th Conference on Neural Information Processing Systems (NeurIPS 2021).

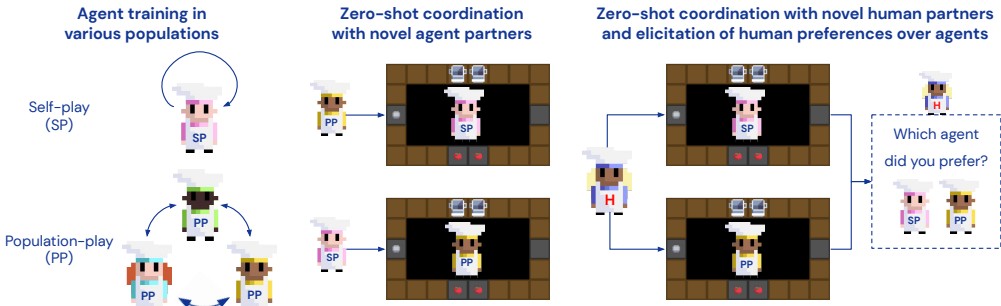

Figure 1: In this work, we evaluate a variety of agent training methods (Section 2) in zero-shot coordination with agents (Section 4). We then run a human-agent collaborative study designed to elicit human preferences over agents (Section 5).

[12]. PP in common-payoff settings naturally encourages agents to play the same way, reducing strategic diversity and producing agents not so different from self-play [24].

Our approach starts with the intuition that the key to producing robust agent collaborators is exposure to diverse training partners. We find that a surprisingly simple strategy is effective in generating sufficient diversity. We train $N$ self-play agents varying only their random seed for neural network initialization. Periodically during training, we save agent "checkpoints" representing their strategy at that point in time. Then, we train an agent partner as the best-response to both the fully-trained agents and their past checkpoints. The different checkpoints simulate different skill levels, and the different random seeds simulate breaking symmetries in different ways. We refer to this agent training procedure as **Fictitious Co-Play (FCP)** for its relationship to fictitious self-play [7, 27, 28, 69].

We evaluate FCP in a fully-observable two-player common-payoff collaborative cooking simulator. Based on the game Overcooked [25], it has recently been proposed as a coordination challenge for AI [12, 50, 70]. State-of-the-art performance in producing agents capable of generalization to novel humans was achieved in [12] via behavioral cloning (BC) of human data. More precisely, BC was used to produce models that can stand in as human proxies during training in simulation, a method we call behavioral cloning play (BCP). We demonstrate that FCP outperforms BCP in generalizing to both novel agent and human partners, and that humans express a significant preference for partnering with FCP over BCP. Our method avoids the cost and potential privacy concerns of collecting human data for training, while achieving better outcomes for humans at test time.

We summarize the novel contributions of this paper as follows:

1. We propose Fictitious Co-Play (FCP) to train agents capable of zero-shot coordination with humans (Section 2.1).
2. We demonstrate that FCP agents generalize better than SP, PP, and BCP in zero-shot coordination with a variety of held-out agents (Section 4.2).
3. We propose a rigorous human-agent interaction study with behavioral analysis and participant feedback (Section 5.1).
4. We demonstrate that FCP significantly outperforms the BCP state-of-the-art, both in task score and in human partner preference (Section 5.2).

## 2 Methods

### 2.1 Fictitious Co-Play (FCP)

Diverse training conditions have been shown to make agents more robust, from environmental variations (i.e. domain randomization [54, 56, 67]) to heterogeneity in training partners [69]. We seek to train agents that are robust partners for humans in common-payoff games, and so extend this line of work to that setting.

One important challenge in collaborating with novel partners is dealing with symmetries [31]. For example, two agents A and B facing each other may move past each other by A going left and B going right, or vice versa. Both are valid solutions, but a good agent partner will adaptively switch between

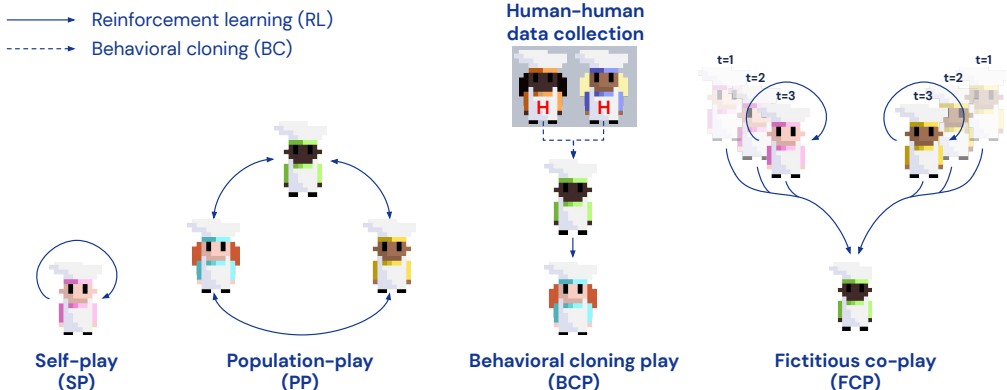

Figure 2: The four agent training methods we evaluate in this work. **Self-play (SP)** where an agent learns with itself, **population-play (PP)** where a population of agents are co-trained together, and **behavioral cloning play (BCP)** where data from human games is used to create a behaviorally cloned agent with which an RL agent is then trained. In our method, **Fictitious Co-Play (FCP)**, $N$ self-play agents are trained independently and checkpointed throughout training. An agent is then trained to best respond to the entire population of SP agents and their checkpoints.

these conventions if a human clearly prefers one over the other. A second important challenge is dealing with variations in skill level. Good agent partners should be able to assist both highly-skilled partners, as well as partners who are still learning.

Fictitious co-play (FCP) is a simple two-stage approach for training agents that overcomes both of these challenges (Figure 2, right). In the first stage, we train a diverse pool of partners. To allow the pool to represent different symmetry breaking conventions, we train $N$ partner agents in self-play. Since these partners are trained independently, they can arrive at different arbitrary conventions for breaking symmetries. To allow the pool to represent different skill levels, we use multiple checkpoints of each self-play partner throughout training. The final checkpoint represents a fully-trained "skillful" partner, while earlier checkpoints represent less skilled partners. Notably, by using multiple checkpoints per partner, this additional diversity in skill incurs no extra training cost.

In the second stage, we train an FCP agent as the best response to the pool of diverse partners created in the first stage. Importantly, the partner parameters are frozen and thus FCP must learn to adapt to partners, rather than expect partners to adapt to it. In this way, FCP agents are prepared to follow the lead of human partners, and learn a general policy across a range of strategies and skills. We call our method "fictitious" co-play for its relationship to fictitious self-play in which competitive agents are trained with past checkpoints (in that case, to avoid strategy cycling) [7, 27, 28, 39, 69].

## 2.2 Baselines and ablations

We compare FCP agents to the three baseline training methods listed below, each varying only in their set of training partners, with the RL algorithm and architecture consistent across all agents:

1. Self-play (SP), where agents learn solely through interaction with themselves.
2. Population-play (PP), where a population of agents are co-trained through random pairings.
3. Behavioral cloning play (BCP), where an agent is trained with a BC model of a human [12].

We also evaluate three variations on FCP to better understand the conditions for its success:

1. To test the importance of including past checkpoints in training, we evaluate an ablation of FCP in which agents are trained only with the converged checkpoints of their partners (FCP$_{-T}$ for "FCP minus time").
2. To test whether FCP would benefit from additional diversity in its partner population, we evaluate an augmentation of FCP in which the population of SP partners varies not just in random seed, but also in architecture (FCP$_{+A}$ for "FCP plus architectural variation").
3. To test whether architectural variation can serve as a full replacement for playing with past checkpoints, we evaluate the combination of both modifications (FCP$_{-T,+A}$).

## 2.3 Environment

Following prior work on zero-shot coordination in human-agent interaction, we study the Overcooked environment (see Figure 3) [12, 13, 38, 50, 70]. We draw particular inspiration from the environment in Carroll et al. [12]. For full details, see Appendix A.

In this environment, players are placed into a gridworld kitchen as chefs and tasked with delivering as many cooked dishes of tomato soup as possible within an episode. This involves a series of sequential high-level actions to which both players can contribute: collecting tomatoes, depositing them into cooking pots, letting the tomatoes cook into soup, collecting a dish, getting the soup, and delivering it. Upon a successful delivery, both players are rewarded equally.

To effectively complete the task, players must learn to navigate the kitchen and interact with objects in the correct order, all while maintaining awareness of their partner's behavior to coordinate with them. This environment therefore presents the challenges of both movement and strategic coordination.

Each player observes an egocentric RGB view of the world, and at every step can perform one of six actions: `stand still`, `move {up, down, left, right}`, `interact`. The behavior of `interact` varies based on the cell which the player is facing (e.g. place tomato on counter).

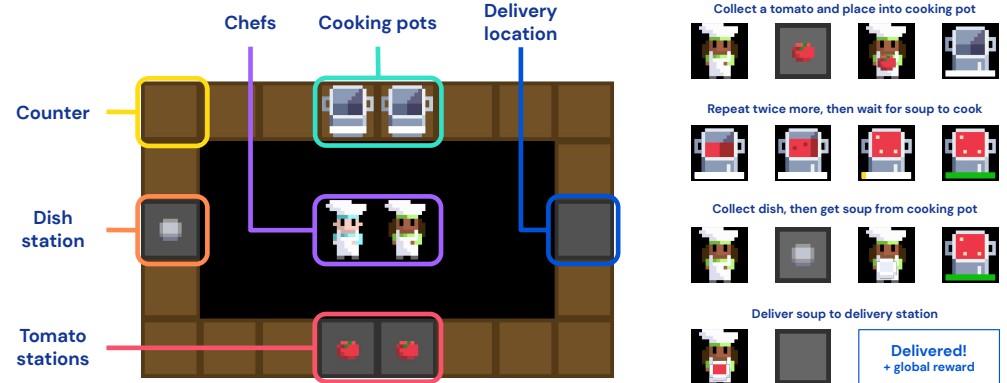

Figure 3: **The Overcooked environment:** a two-player common-payoff game in which players must coordinate to cook and deliver soup.

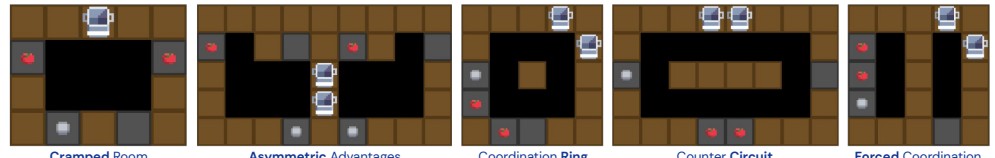

Figure 4: **Layouts:** the kitchens which agents and humans play in, each emphasizing different coordination strategies. Highlighted in bold are the terms used to refer to each in the rest of this paper.

## 2.4 Implementation details

Here we highlight several key implementation details for our training methods. For full details, including the architectures, hyperparameters, and compute used, please see Appendix B.

For our reinforcement learning agents, we use the V-MPO [65] algorithm along with a ResNet [26] plus LSTM [29] architecture which we found led to optimal behavior across all layouts. Agents are trained using a distributed set of environments running in parallel [17], each sampling two agents from the training population to play together every episode.

Both PP and FCP are trained with a population size of $N = 32$ agents which are sampled uniformly. For FCP, we use 3 checkpoints for each agent, therefore incurring no additional training burden: (1) at initialization (i.e. a low-skilled agent), (2) at the end of training (i.e. a fully-trained expert agent), and (3) at the middle of training, defined as when the agent reaches 50% of its final reward (i.e. an average-skilled agent). When varying architecture for the training partners of the $FCP_{+A}$ and

$FCP_{-T,+A}$ variants, we vary whether the partners use memory (i.e. LSTM vs not) and the width of their policy and value networks (i.e. 16 vs 256). In total, we train 8 agents for each of the 4 combinations, leaving the total population size of $N = 32$ unchanged, ensuring a fair comparison.

To train agents via behavioral cloning [58], we use the open-source Acme [30] to learn a policy from human gameplay data. Specifically, we collected 5 human-human trajectories of length 1200 time steps for each of the 5 layouts, resulting in 60k total environment steps. We divide this data in half and train two BC agents: (1) a partner for training a BCP agent, and (2) a "human proxy" partner for agent-agent evaluation. Following Carroll et al. [12], we use a set of feature-based observations for the agents (as opposed to RGB) and generate comparable results: performance is higher on 3 layouts (asymmetric, cramped, and ring) but poorer on the other 2 (circuit and forced).

## 3  Related work

**Ad-hoc team play**    There is a large and diverse body of literature on ad-hoc team-play [5, 66], also known as zero-shot coordination [31]. Prior work based in game-theoretic settings has suggested the benefits of planning [71], online learning [51], and novel solution concepts [2], to name a few examples. More recently, multi-agent deep reinforcement learning has provided the tools to scale to more complex gridworld or continuous control settings, leading to work on hierarchical social planning [36], adapting to existing social conventions [40, 62], trajectory diversity [45], and theory of mind [14]. Ad-hoc team-play among novel agent partners is also an object of active study in the emergent communication literature [10, 11, 43]. This prior work has tended to focus on generalization to held-out agent partners as a proxy for human co-players.

Collaborative play with novel humans has been evaluated more actively in the context of training agent assistants; see for instance [57, 68]. To our knowledge, our FCP agents represent the state-of-the-art in coordinating with novel human partners on an equal footing of capabilities in a rich gridworld environment, as measured by the challenge tasks in Carroll et al. [12].

**Diversity in multi-agent reinforcement learning**    In multi-agent reinforcement learning, agents that train with behaviorally diverse populations of game partners tend to demonstrate stronger performance than their self-play counterparts. For example, across a range of multi-agent games, generalization to held-out populations can be improved by training larger and more diverse populations [13, 42, 50]. In mixed-motive settings, cooperation among agents can be encouraged through social diversity, such as in player preferences and rewards [3, 47, 49]. Similarly, competitiveness can be optimized through selective matchmaking between increasingly diverse agents [24, 39, 69].

Despite the increased focus on improving multi-agent performance, evaluation has typically been constrained to agent-agent settings. High-performing agents have infrequently been evaluated with humans, particularly in non-competitive domains [16]. We add to this growing literature, showing that training with diversity is a powerful approach for effective human-agent collaboration.

**Human-agent interaction**    In recent years, increased attention has been directed toward designing machine learning agents capable of collaborating with humans [41, 57, 68, 72] (see also [16] for a broader review on Cooperative AI). Tylkin et al. [68] is particularly notable in also demonstrating that partially trained agents can be useful learning targets for human helpers, although in a different domain (cooperative Atari). Our method, FCP, can be seen as extending theirs by training with multiple "skill levels" and random seeds, rather than just one, which we demonstrate to be crucial to our agents' performance (Tables 1 and 2 and Figure 7b).

A key preceding entry in this research area is Carroll et al. [12], who similarly investigated human-agent coordination in Overcooked. We use their method (BCP) as a baseline throughout our experiments (Section 2.2). Relative to BCP, our approach removes the need for the expensive step of human data collection for agent training. Furthermore, through our novel human-agent experimental design, we go beyond objective performance metrics to compare the subjective preferences that agents generate. For a detailed comparison of methods and results, see Appendix E.

## 4  Zero-shot coordination with agents

In this section, we evaluate our FCP agent, its ablations, and the baselines with held-out agents.

### 4.1 Evaluation method: collaborative evaluation with agent partners

Our primary concern in this work is generalization to novel *human* partners (as investigated in Section 5). However, just as collecting human-human data for behavioral cloning is expensive, so too is evaluating agents with humans. Consequently, we instead use generalization to held-out *agent* partners as a cheap proxy of performance with humans. This is then used to guide our model selection process, allowing us to be more targeted with the agents we select for our human-agent evaluations.

We evaluate with three held-out populations:

1. A BC model trained on human data, $H_{\text{proxy}}$, intended as a proxy of generalization to humans, as done by Carroll et al. [12].
2. A set of self-play agents varying in seed, architecture, and training time (specifically, held-out seeds of the $N = 32$ partners trained for the $\text{FCP}_{+A}$ agent; see Section 2.4). These are intended to test generalization to a diverse yet still skillful population.
3. Randomly initialized agents intended to test generalization to low-skill partners.

For all results, we report the average number of deliveries made by both players within an episode, aggregated across the 5 different layouts from Figure 4 (with the per-layout results reported in Appendix C.2). We estimate mean and standard deviation across 5 random seeds. For each seed, we evaluate the agent with all members of the held-out population for 10 episodes per agent-partner pair.

### 4.2 Results

**Finding 1: FCP significantly outperforms all baselines**

To begin, we compare our FCP agent and the baselines when partnered with the three held-out populations introduced above. As can be seen in Figure 5, FCP significantly outperforms all baselines when partnered with all three held-out populations. Notably, it performs better than BCP with $H_{\text{proxy}}$, even though BCP trains with such a model and FCP does not. Similar to Carroll et al. [12], we find that BCP significantly outscores SP.

When paired with a randomly initialized partner which behaves suboptimally, we see an even greater difference between FCP and the baselines. Given that FCP is trained with non-held-out versions of such agents, it may not be surprising that it does so well with partners that behave poorly. However, what is surprising is how brittle the other training methods are. This suggests that they may not perform well with humans who are not highly skilled players, which we will see in Section 5.

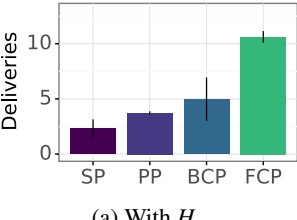
(a) With $H_{\text{proxy}}$.

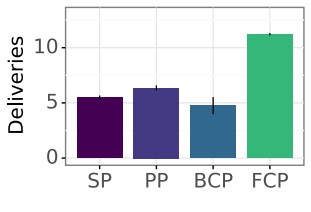
(b) With diverse SP agents.

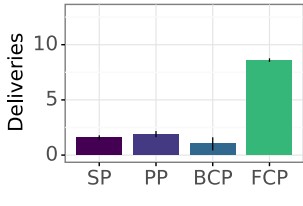
(c) With random agents.

Figure 5: **Agent-agent collaborative evaluation:** Performance of each agent when partnered with each of the held-out populations (Section 4.1) in episodes of length $T = 540$. Importantly, FCP scores higher than all baselines with a variety of test partners. Error bars represent standard deviation over five random training seeds. Plots aggregate data across kitchen layouts; results calculated by individual layout can be found in Appendix C.2.

**Finding 2: Training with past checkpoints is the most beneficial variation for performance**

Next, we investigate how the different training partner variations influence FCP's performance. In particular, we separately ablate the past checkpoints ($T$) and architecture ($A$) variations, evaluating them with the same partners as in Figure 5. The results of this evaluation are presented in Table 1.

Comparing the FCP and $\text{FCP}_{-T}$ columns, we see that removing past checkpoints from training significantly reduces performance. Comparing the FCP and $\text{FCP}_{+A}$ columns, we see that adding architectural variation to the training population offers no improvement over training with past

| Partner | FCP | FCP$_{-T}$ | FCP$_{+A}$ | FCP$_{-T,+A}$ |
|---|---|---|---|---|
| $H_{\text{proxy}}$ | $10.6 \pm 0.5$ | $4.7 \pm 0.4$ | $9.9 \pm 0.6$ | $7.0 \pm 0.8$ |
| Diverse SP | $11.2 \pm 0.1$ | $6.9 \pm 0.1$ | $11.1 \pm 0.4$ | $8.6 \pm 0.4$ |
| Random | $8.6 \pm 0.2$ | $1.0 \pm 0.1$ | $8.4 \pm 0.4$ | $3.2 \pm 0.5$ |

Table 1: **Ablation results:** Performance of each variation of FCP – training with past partner checkpoints ($T$ for time) and adding partner variation in architecture ($A$). Scores are mean deliveries with standard deviation over 5 random seeds. Notably, we find that the inclusion of past checkpoints is essential for strong performance (FCP > FCP$_{-T}$), and additionally including architectural variation does not improve performance (FCP $\approx$ FCP$_{+A}$). However, architectural variation is better than no variation, improving performance when past checkpoints are not available (FCP$_{-T,+A}$ > FCP$_{-T}$).

checkpoints. However, comparing the FCP$_{-T}$ and FCP$_{-T,+A}$ columns, we see that without training with past checkpoints, architectural variation in the population does improve performance.

## 5 Zero-shot coordination with humans

Ultimately, our goal is to develop agents capable of coordinating with novel human partners. In this section, we run an online study to evaluate our FCP agent and the baseline agents in collaborative play with human partners.

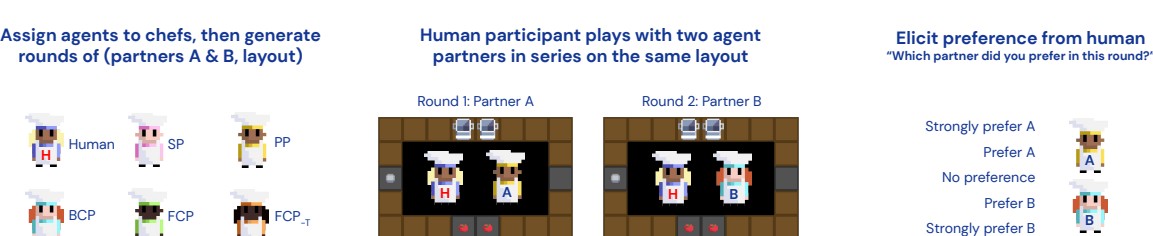

Figure 6: **Human-agent collaborative study:** For our human-agent collaboration study, we recruited participants online to play games with FCP and baseline agents. Participants played a randomized sequence of episodes with different agent partners and kitchen layouts. After every two episodes, participants reported the direction and strength of their preference between their last two partners.

### 5.1 Evaluation method: collaborative evaluation with human participants

To test how effectively FCP's performance generalizes to human partners, we recruited participants from Prolific [18, 55] for an online collaboration study ($N = 114$; 37.7% female, 59.6% male, 1.8% nonbinary; median age between 25–34 years). We used a within-participant design for the study: each participant played with a full cohort of agents (i.e. generated through every training method). This design allowed us to evaluate both objective performance as well as subjective preferences.

Participants first read game instructions and played a short tutorial episode guiding them through the dish preparation sequence (see Appendix D.1.1 for instruction text and study screenshots). Participants then played 20 episodes with a randomized sequence of agent partners and kitchen layouts. Episodes lasted $T = 300$ steps (1 minute) each. After every two episodes, participants reported their preference over the agent partners from those episodes on a five-point Likert-type scale. After playing all 20 episodes, participants completed a debrief questionnaire collecting standard demographic information and open-ended feedback on the study. Our statistical analysis below primarily relies upon the repeated-measures analysis of variance (ANOVA) method. See Appendix D for additional details of our study design and analysis, including independent ethical review.

### 5.2 Results

**Finding 1: FCP coordinates best with humans, achieving the highest score across maps**

To begin, we compare the objective team performance supported by our FCP and baseline agents. The strong FCP performance observed in agent-agent play generalizes to human-agent collaboration:

the FCP-human teams significantly outperform all other agent-human teams, achieving the highest average scores across maps, every $p < 0.001$ (Figure 7a), while performing as well as or better than the other teams on each individual map (see Appendix D.3). Echoing the results from our agent-agent ablation experiments (Table 1), the inclusion of past checkpoints in training proves critical to FCP's strong performance, $p < 0.001$ (Figure 7b). Similar to Carroll et al. [12], we find that BCP outscores SP when collaborating with human players, $p < 0.001$.

**Finding 2: Participants prefer FCP over all baselines**

FCP's strong collaborative performance carries over to our participants' subjective partner preferences. Participants expressed a significant preference for FCP partners over all other agents, including BCP, with every $p < 0.05$ (Figure 7c). Notably, while human-BCP and human-PP teams did not significantly differ in their completed deliveries, participants reported significantly preferring BCP over PP, $p = 0.003$, highlighting the informativeness of our subjective analysis.

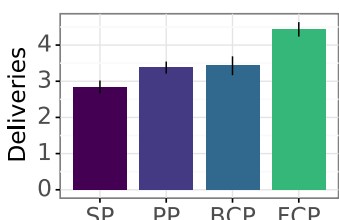

(a) Number of deliveries by partner (FCP and baselines).

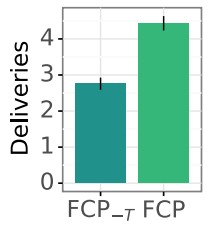

(b) Number of deliveries by partner (FCP and $FCP_{-T}$).

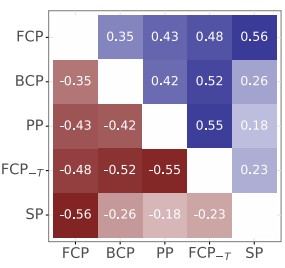

(c) Participant preference for row partner over column partner.

Figure 7: **Human-agent collaborative evaluation**: Evaluation and preference metrics from human-agent play in episodes of length $T = 300$. Error bars represents 95% confidence intervals, calculated over episodes. Plots aggregate data across kitchen layouts; results calculated by individual layout can be found in Appendix D.3.

### 5.3 Exploratory behavioral analysis

To better understand how the human-agent scores and preferences may have arisen, here we analyze the resulting action trajectories of each human and agent player in our experiment.

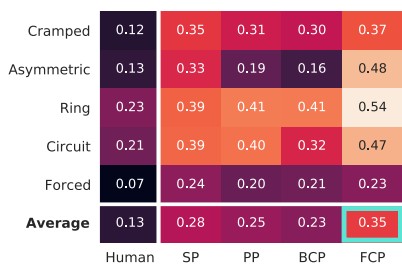

(a) Proportion of episode spent moving.

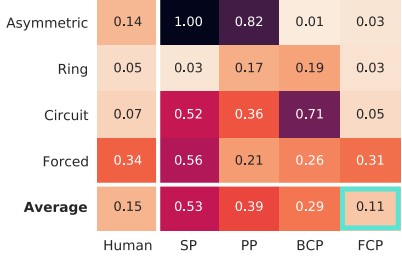

(b) Differences in pot preference.

Figure 8: **Behavioral analysis:** (a) FCP is able to move most frequently (35% of the time), corresponding to the best movement coordination with human partners. (b) FCP exhibits the most equal preferences over cooking pots (0.11 difference), aligning with human preferences. Values are calculated as the absolute difference in preferences between the two pots; 1 indicates that the player only uses one of the two available pots, while 0 indicates that the player uses both pots equally.

**Finding 1: FCP exhibits the best movement coordination with humans**

First, we investigate how much each player moves in an episode (Figure 8a), where moving in a higher fraction of timesteps may suggest fewer collisions and thus better coordination with a partner. Notably, we observe two results: (1) humans rarely move, a behavior which is out-of-distribution for typical training methods (e.g. SP, PP) but is seen in the training distribution for BCP and FCP.

(2) FCP moves the most on all layouts other than Forced, suggesting it is better at coordinating its movement strategy with its partner. This result was also reported by human participants, for example: "I noticed that some of my partners seemed to know they needed to move around me, while others seemed to get 'stuck' until I moved out of their way" (see Appendix D for more examples).

**Finding 2: FCP's preferences over cooking pots aligns best with that of humans**

Next, we investigate whether there was a preference for a specific cooking pot in the layouts which included two cooking pots (Figure 8b). To do this, we calculate the difference in the number of times each pot was used by each player, where a high value indicates a strong preference for one pot and a low value indicates more equal preference for the two pots.

As can be seen in the FCP column, our agent typically has the most aligned preferences with that of humans ($0.11$ for FCP to $0.14$ for humans). Behaviorally speaking, this means that our agent prefers one cooking pot over the other $55.5\%$ of the time (i.e. a $0.11$ point difference). In contrast, all other agents have a strong preference for a single pot. This is a non-adaptive strategy which generalizes poorly to typical human behavior of using both pots, leading to worse performance.

## 6 Discussion

**Summary**   In this work, we investigated the challenging problem of zero-shot collaboration with humans without using human data in the training pipeline. To accomplish this, we introduced Fictitious Co-Play (FCP) – a surprisingly simple yet effective method based on creating a diverse set of training partners. We found that FCP agents scored significantly higher than all baselines when partnered with both novel agent and human partners. Furthermore, through a rigorous human-agent experimental design, we also found that humans reported a strong subjective preference to partnering with FCP agents over all baselines.

**Limitations and future work**   Our method currently relies on the manual process of initially training and selecting a diverse set of partners. This is not only time consuming, but also prone to researcher biases that may negatively influence the behavior of the created agents. Additionally, while we found FCP with a partner population size of $N = 32$ sufficient here, for more complex games, FCP may require an unrealistically large partner population size to represent sufficiently diverse strategies. To address these concerns, methods for automatically generating partner diversity for common-payoff games may be important. Possibilities include adaptive population matchmaking as been used in competitive zero-sum games [69], as well as auxiliary objectives that explicitly encourage behavioral diversity [19, 45, 46].

Our method requires a known and fixed reward function. We also focus on one domain in order to compare with prior work which has argued that human-in-the-loop training is necessary. Consequently, the resulting agents are only designed to adaptively collaborate on a single task, and not to infer human preferences in general [1, 33, 59]. Moreover, if a task's reward function is poorly aligned with how humans approach the task, our method may well produce subpar partners, as would any method without access to human data. Thus, additional domains and tasks should be studied to better understand how our method generalizes. Targeted experiments to test specific forms of generalization may be especially helpful in this regard [38], as could approaches that procedurally generate environment layouts requiring diverse solutions [22].

Finally, it may be possible to produce even stronger agent assistants by combining the strengths of FCP (i.e. diversity) and BCP (i.e. human-like play). Indeed, Knott et al. [38] recently demonstrated that modifying BCP to train with *multiple* BC partners produces more robust collaboration with held-out agents, a finding that would be interesting to test with human partners.

**Societal impact**   A challenge for this line of work is ensuring agent behavior is aligned with human values (i.e. the AI value alignment problem [23, 59]). Our method has no guarantees that the resulting policy aligns with the preferences, intentions, or welfare of its potential partners. It likewise does not exclude the possibility that the target being optimized for is harmful (e.g. if the agent's partner expresses preferences or intentions to harm others). This could therefore produce negative societal effects either if training leads to poor alignment or if agents are optimized for harmful metrics.

One potential strategy for mitigating these risks is the use of human preference data [15]. Such data could be used to fine-tune and filter trained agents before deployment, encouraging better alignment with human values. A key question in this line of research is how human preference data should be

aggregated—or selected, in the case of expert preferences—when our aim is to create socially aligned agents (i.e. agents that are sufficiently aligned for everyone). Relatedly, targeted research on human beliefs and perceptions of AI [48], and how they steer human-agent interaction, would help inform agent design for positive societal impact. For instance, developers could incorporate specific priors into agents to reinforce tendencies for fair outcomes [20, 32].

**Conclusion** We proposed a method which is both effective at collaborating with humans and simple to implement. We also presented a rigorous and general methodology for evaluating with humans and eliciting their preferences. Together, these establish a strong foundation for future research on the important challenge of human-agent collaboration for benefiting society.

## Acknowledgements

The authors would like to thank Mary Cassin for creating the game sprite art; Rohin Shah, Thore Graepel, and Iason Gabriel for feedback on the draft; Lucy Campbell-Gillingham, Tina Zhu, and Saffron Huang for support in evaluating agents with humans; and Max Kleiman-Weiner, Natasha Jaques, Marc Lanctot, Mike Bowling, and Dan Roberts for useful discussions.

## Funding disclosure

This work was funded solely by DeepMind. The authors declare no competing interests.

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
