# A Environment details

## A.1 Gameplay

Players are placed in a gridworld environment containing cooking pots, tomato stations, dish stations, delivery locations, and empty counters. Players can move around and interact with these objects. By sequencing certain object interactions, players can pick up and deposit items. Each player (and each counter) can only hold one item at a time.

The objective of each episode is for the players to deliver as many tomato soup dishes as possible to delivery locations. In order to create a tomato soup dish, players must pickup tomatoes and deposit them into the cooking pot. Once there are three tomatoes in the cooking pot, it begins to cook for 20 steps. After 20 steps, the soup is fully cooked. Cooking progress for the soup is tracked by a loading bar overlaying the cooking pot. The loading bar increments over the cooking time and then turns green when the soup is ready for collection.

When the tomato soup is ready for collection, a player holding an empty dish can interact with the cooking pot to pick up the soup. The player can then deliver the soup by interacting with a delivery station while holding the completed dish. A successful delivery rewards both players and removes the dish from the game.

## A.2 Observations

Importantly, our agent and human players observe different views of the environment. We found that agents trained better and were more robust using egocentric observations (Figure 9a). However, humans players found this disorienting and so observed the world from a static top-down perspective of the environment (Figure 9b).

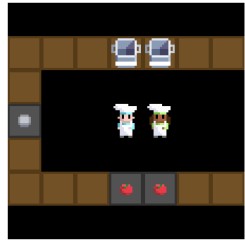 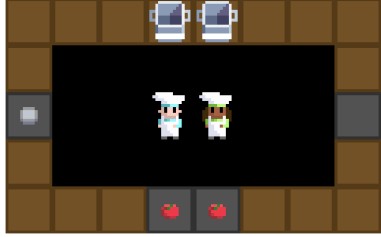

(a) Agent observation (cyan).         (b) Human observation.

Figure 9: Example observations: (a) Agent players observe a $56 \times 56 \times 3$ egocentric view of the environment (i.e., $7 \times 7$ cells with a $8 \times 8 \times 3$ sprite in each cell). (b) Human players observe the full layout from a top-down perspective.

## A.3 Actions

Players can take one of the following eight actions each step:

1. `No-op`: The player stays in the same position.
2. `Move up`: Moves the player up one cell.
3. `Move down`: Moves the player down one cell.
4. `Move left`: Moves the player left one cell.
5. `Move right`: Moves the player right one cell.
6. `Interact`: The player interacts with the cell that they are facing.

The outcome of the `Interact` action depends on the current item held by the player (none, empty dish, tomato, or soup), as well as the type of object which they are facing (counter, cooking pot, tomato station, dish station, or delivery station). Depending on these two conditions, the player will either deposit the held item to the object or pick up an item from the object.

## A.4 Rewards

Players receive a shared $+20$ reward for every soup they deliver through the delivery station. As a result, they are incentivized to deliver as many soups as possible within an episode. To help scaffold learning, players also receive $+1$ reward each time they deposit a tomato into the cooking pot.

## A.5 Layouts

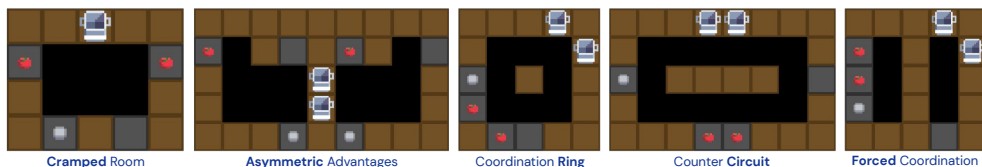

Figure 10: **Layouts:** the kitchens which agents and humans play in, each emphasizing different coordination strategies. Highlighted in bold are the terms used to refer to each in the paper.

- **Cramped:** A tight layout requiring significant movement coordination between the players in order to avoid being blocked by each other.

- **Asymmetric:** A two-room layout with an agent in each. In the left room, the tomato station is far away from the cooking pots while the delivery location is close. In the right room, the tomato station is next to the cooking pots while the delivery station is far. This presents an asymmetric advantage of responsibilities for optimally creating and delivering soups.

- **Ring:** A layout with two equally successful movement strategies – (1) both players moving clockwise, and (2) both players moving anti-clockwise. If players do not coordinate, they will block each other's movement.

- **Circuit:** Players are able to cook and deliver soups by themselves through walking around the entire circuit. However, there exists a more optimal coordinated strategy whereby players pass tomatoes across the counter. Additionally, there are the clockwise and anti-clockwise strategies as in the Ring layout.

- **Forced:** One player is in the left room and second player is in the right room. Consequently, both players are forced to work together in order to cook and deliver soup. The player in the left room can only pass tomatoes and dishes, while the player on the right can only cook the soup and deliver it (using the items provided by the first player).

# B  Agent details

## B.1  Fictitious Co-Play

We provide pseudocode for Fictitious Co-Play (FCP) in Algorithm 1. Unless otherwise stated, all FCP results are with a partner population size of $N = 32$. We used a checkpoint frequency $n_c$ of $1 \times 10^7$ steps, resulting in a total of 100 checkpoints saved per training run of $1 \times 10^9$ steps. After the first stage of training partners, we filter checkpoints (i.e. $F$) down to three for each partner: the first checkpoint (i.e. a randomly initialized agent), the last checkpoint (i.e. a fully trained agent),

and the remaining checkpoint that achieves closest to half of the reward of the last checkpoint (i.e. a half-trained agent).

---

**Algorithm 1:** Fictitious Co-Play (FCP)

---

**Input:** Number of partners $N$, checkpoint frequency $n_c$, checkpoint filter $F$
`// Stage 1: train diverse partner population`
partners = []
**for** $i = 1$ **to** $N$ **do**
    Initialize agent $i$.
    $n = 0$ `// step count`
    **while** *not converged* **do**
        Update agent $i$ in self-play.
        $n \mathrel{+}= 1$
        **if** $n \mod n_c = 0$ **then**
            Add frozen agent $i$ checkpoint to partners.
`// Stage 2: train FCP agent`
Filter partners with $F$.
Initialize FCP agent.
**while** *not converged* **do**
    Sample partner from partners.
    Update FCP in co-play with partner.

---

## B.2 Training settings

Agents are trained using a distributed set of environments running in parallel. Each agent is trained using one GPU on $N \times 200$ environments, where $N$ is the number of agents being trained in the population. Agents are trained for $1 \times 10^9$ environment steps which takes between three and eight days depending on the size of the training population. As the environment involves two players, each one samples with replacement from the training population of agents every episode.

## B.3 Deep reinforcement learning (DRL)

### B.3.1 Architecture and hyperparameters

We used the following architecture for V-MPO [65]. The agent's visual observations were first processed through a 3-section ResNet used in McKee et al. [50]. Each section consisted of a convolution and $3 \times 3$ max-pooling operation (stride 2), followed by residual blocks of size 2 (i.e., a convolution followed by a ReLU nonlinearity, repeated twice, and a skip connection from the input residual block input to the output). The entire stack was passed through one more ReLU nonlinearity. All convolutions had a kernel size of 3 and a stride of 1. The number of channels in each section was (16, 32, 32).

The resulting output was then concatenated with the previous action and reward of the agent, and processed by a single-layer MLP with 256 hidden units. This was followed by a single-layer LSTM with 256 hidden units (unrolled for 100 steps), and then a separate single-layer MLP with 256 hidden units to produce the action distribution. For the critic, we used a single-layer MLP with 256 hidden units followed by PopArt normalization (which leads to stronger multi-agent performance [50]).

To train the agent, we used a discount factor of 0.99, batch sizes of 16, and the Adam optimizer (learning rate of 0.0001). We configured V-MPO with a target network update period of 100, $k = 0.5$, and an epsilon temperature of 0.1. For PopArt normalization, we used a scale lower bound of $1\mathrm{e}{-2}$, an upper bound of $1\mathrm{e}6$, and a learning rate of $1\mathrm{e}{-3}$.

### B.3.2 Training curves

## B.4 Behavioral cloning (BC)

### B.4.1 Architecture and hyperparameters

We trained a single network consisting of five sub-networks (1 per layout), each being a three-layer MLP with 256 hidden units per layer. The network was then trained using all of the recorded human

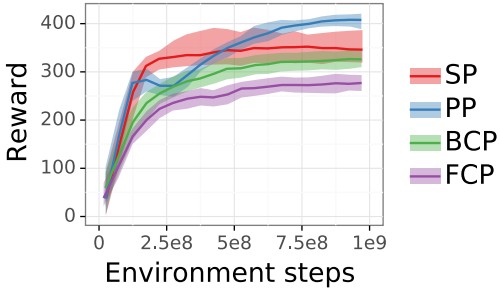

Figure 11: Training curves for agents evaluated. Median and 75% confidence interval over 5 seeds. Note that agents vary in the partners they train with, so the scores should not be directly compared to each other (e.g. FCP training partners include randomly initialized agents).

trajectories (i.e. on all layouts), with the sub-network conditioned on the trajectory's layout feature. We used a batch size of 256 and and the Adam optimizer (learning rate = 0.0003).

### B.4.2 Feature-based observations

As learning from RGB observations requires a significant amount of human data, we instead opted to handcraft a set of features for our BC agent to learn from. In particular, we used the following features: player position and orientation, current held item (as a one-hot vector), the relative position of the other player, the state of each cooking pot (as a one-hot vector of empty, 1 tomato, 2 tomato, 3 tomatoes, and cooked), faced cell is empty, each adjacent cell is empty (length 4), and the relative distance to each object (tomato, dish, soup, and delivery location). For our behavioral cloning, we also included the layout name.

### B.4.3 Training details

We collected 5 human-human trajectories of length 1200 timesteps for each of the 5 layouts, resulting in 60,000 total environment steps. We then divided this data in half to train two BC agents: (1) a partner for training a BCP agent ($H_{\text{partner}}$), and (2) a "human proxy" partner for agent-agent evaluation ($H_{\text{proxy}}$).

$H_{\text{partner}}$ was trained for 32,000 gradient steps (273 epochs) and $H_{\text{proxy}}$ was trained for 61,000 gradient steps (520 epochs). Training time was based on peak reward in self-play, analogous to early stopping.

## C    Zero-shot coordination with agents

### C.1    Evaluation details

For each of SP, BCP, and FCP, we trained 5 random seeds per agent. For PP, we trained a single 32-agent population and chose the first 5 agents for evaluation. All results are averaged across these seeds.

For the held-out evaluation populations, the "diverse SP" group consisted of 60 self-play agents varying in random seed (5 seeds), architecture (4 variants: LSTM vs feedforward, policy and value network widths of 16 vs 256), and training time (3 variants: randomly initialized, half-trained, and fully-trained). The architectural variation is the same type used for the $FCP_{+A}$ agent in Table 1, while the training time variation is the same used for producing partners for FCP. The random agents population were a subset of the diverse SP group, consisting of the 5 seeds of randomly initialized agents. Finally, $H_{\text{proxy}}$ was the BC agent trained on the held-out half of human-human trajectories.

For each pair of agents consisting of one random seed of an agent to be evaluated and one random seed of an agent from a target evaluation population, we played 10 games of length $T = 540$ steps for each layout.

## C.2 Additional results

**Per-layout results**

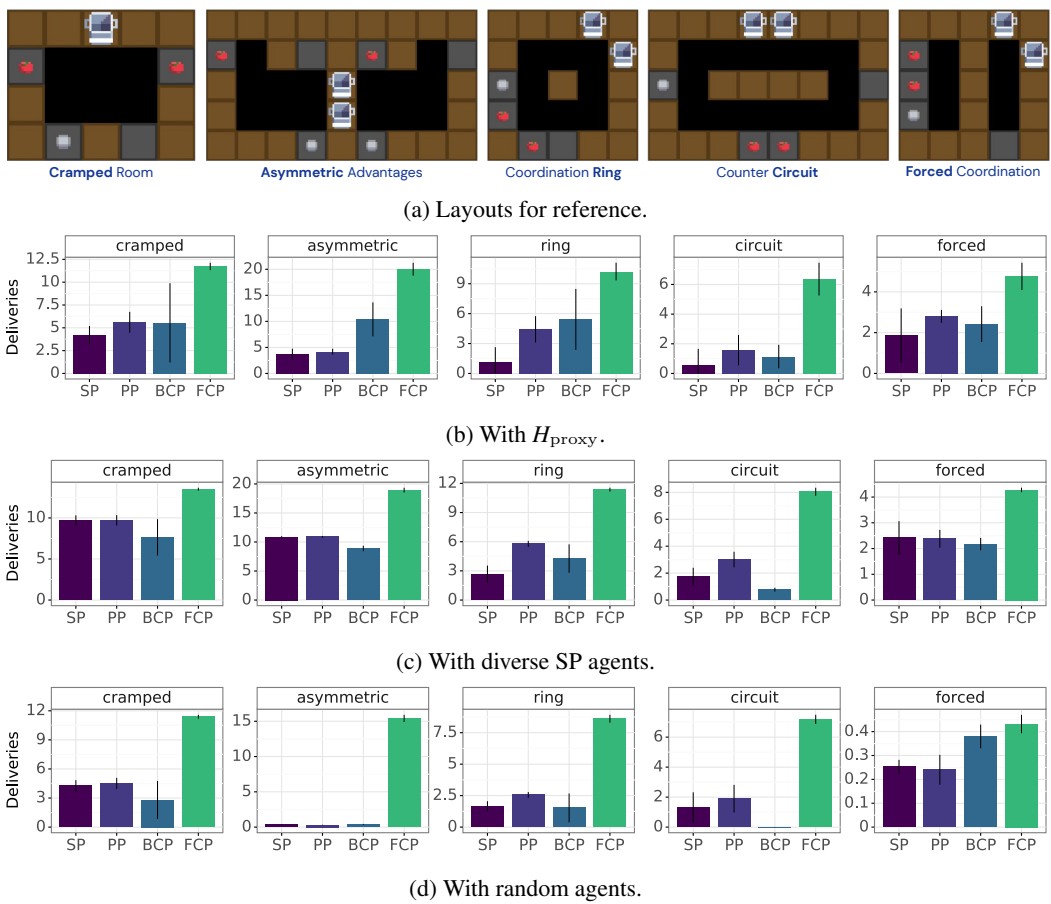

(a) Layouts for reference.

(b) With $H_{\text{proxy}}$.

(c) With diverse SP agents.

(d) With random agents.

Figure 12: **Agent-agent collaborative evaluation, per-layout:** Performance of each agent when partnered with each of the held-out populations in episodes of length $T = 540$. Error bars represent standard deviation over five random training seeds. FCP outperforms all baselines on every map with every partner population.

**Influence of population size on performance**

As we create a training population to train our agent, a natural question to ask is how the size of that population influences the performance of the agent. In Table 2, we present the results of varying $N$ from 4 to 128.

| $N$ | Deliveries |
|-----|------------|
| 4 | 8.4 (0.3) |
| 8 | 9.1 (0.4) |
| 16 | 10.2 (0.4) |
| 32 | 10.6 (0.5) |
| 64 | 10.4 (0.3) |
| 128 | 10.8 (0.6) |

Table 2: Performance of FCP with $H_{\text{proxy}}$, as a function of number of training partners ($N$): larger populations lead to stronger agents and more deliveries. The right column presents mean deliveries with standard deviation over 5 random seeds in parentheses.

We observe a consistent increase in performance as $N$ increases, plateauing around $N = 32$ training partners. This supports the findings of prior work that larger populations are typically stronger, to a point [38, 42, 50]. Consequently, we selected $N = 32$ as our population size across all experiments.

# D  Zero-shot coordination with humans

## D.1  Experimental design

To test how effectively the FCP and baseline agents collaborate with humans in a zero-shot setting, we recruited $N = 114$ participants from Prolific, an online participant recruitment platform [18, 55]. Inclusion criteria were residence in the United States, a minimum approval rate of 95% on prior Prolific studies, and a minimum of 20 prior approved studies.

The study consisted of multiple tutorial pages explaining the game rules and dynamics (Figures 13 and 14), a single-player practice episode (Figure 15), multiple two-player episodes with agent partners (Figures 16-18), and a debrief questionnaire collecting open-ended feedback and demographic information. Each participant played multiple two-player episodes on different layouts. By the end of the study, each participant had collaborated with agents generated through every training method (i.e., the study had a within-participant design). We incentivized game performance: participants earned $0.10 for each dish served, resulting in a cumulative performance bonus at the end of the study. Study sessions lasted 31.2 minutes on average, with a compensation base of $4.00 and an average bonus of $6.74.

DeepMind's independent research ethics committee conducted ethical review for the project and offered a favorable opinion for the study protocol (#19/001), indicating that it presented minimal risk to participants. All participants provided informed consent for the study.

Each participant played with a randomized sequence of agent partners and game layouts. Specifically, we first randomized the order of the five layouts. Layouts were repeated four times in the sequence, so that each participant played 20 episodes total (four episodes on each layout). To generate the sequence of agents that plays in the four episodes for each layout, we sampled four agents without replacement.

### D.1.1 Screenshots

Here we include screenshots of our human-agent collaborative study:

1. Instruction and tutorial screens (Figures 13 and 14).
2. Playing a solo practice episode (Figure 15).
3. Instructions on the episodes with a partner (Figure 16).
4. Playing episode 1 with Partner A (Figure 17).
5. Playing episode 2 with Partner B (Figure 18).
6. Preference elicitation between Partners A and B (Figure 19).
7. Repeat steps 4-6 for 18 more episodes.

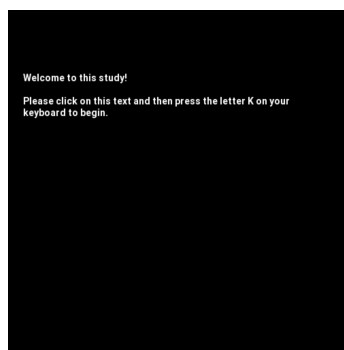
Screen 1: Welcome participants to the experiment.

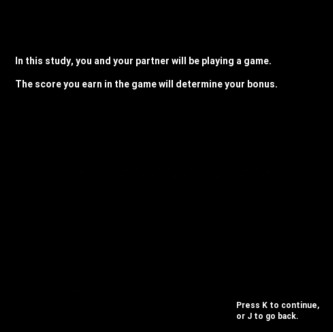
Screen 2: Explain the keyboard controls.

Screen 3: Provide overview of the experiment and bonus.

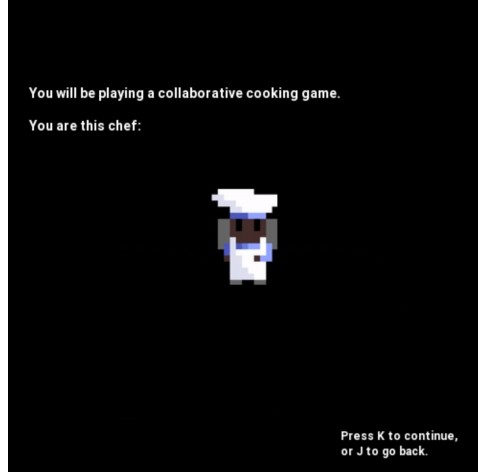
Screen 4: Introduce the participant's controllable chef.
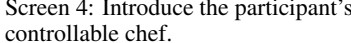

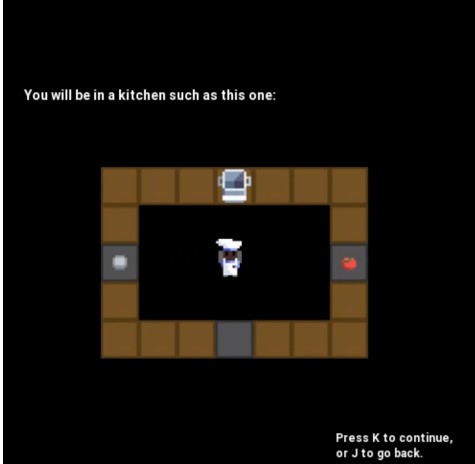
Screen 5: Introduce a generic kitchen layout.

Figure 13: Screenshots of tutorial and instruction screens.

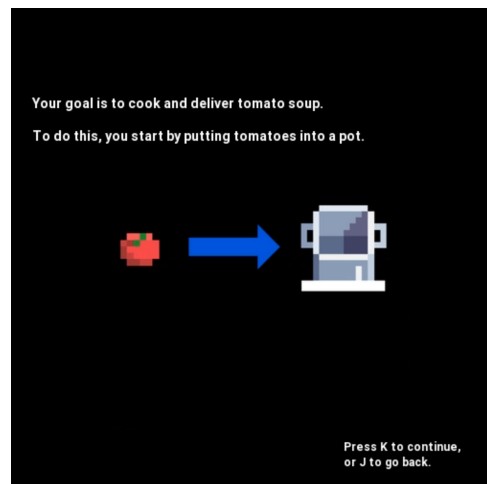

Screen 6: Explain the goal of the game.

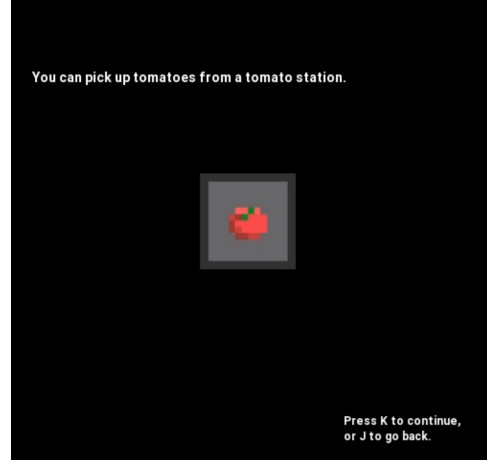

Screen 7: Explain how to collect tomatoes.

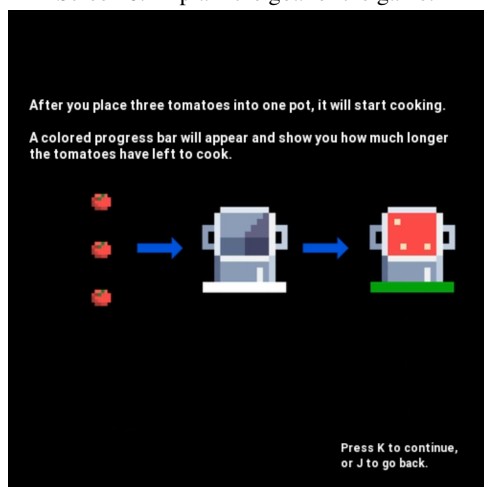

Screen 8: Explain how to cook soup.

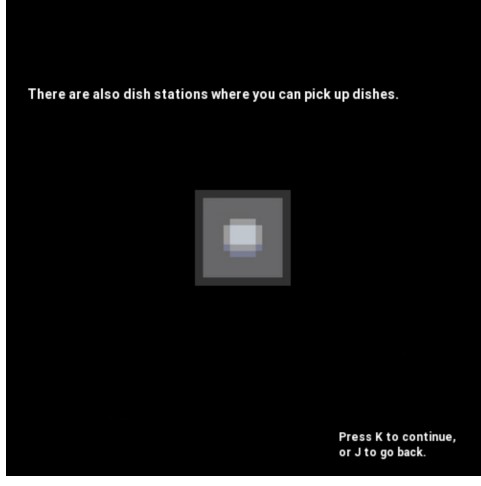

Screen 9: Explain how to collect dishes.

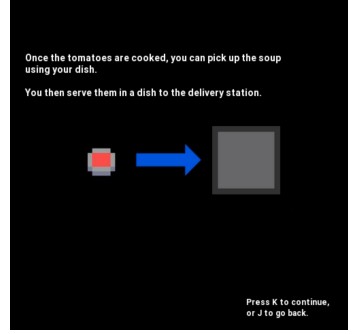

Screen 10: Explain how to deliver soup.

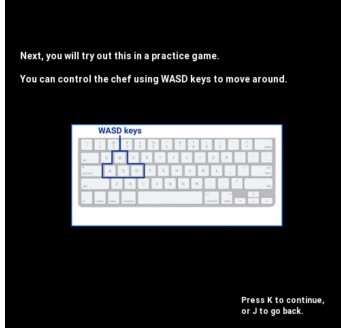

Screen 11: Explain the movement controls for the chef.

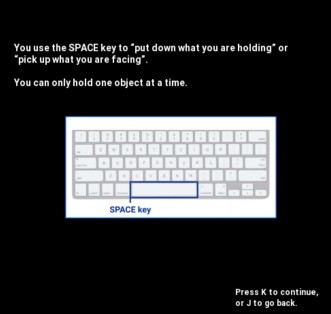

Screen 12: Explain the chef's `interact` action.

Figure 14: Screenshots of tutorial and instruction screens.

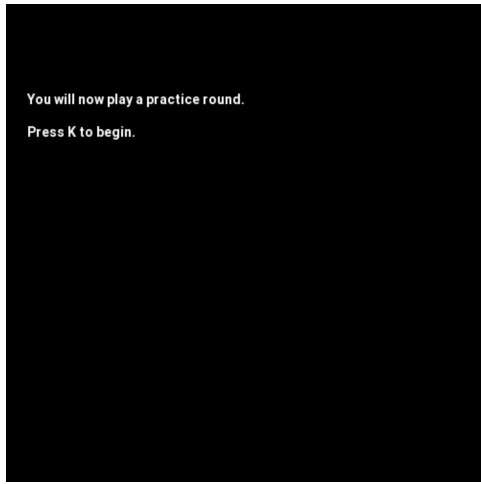

(a) Screen 13: Introduce the practice episode.

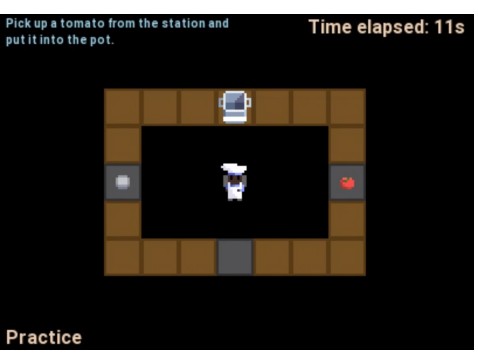

(b) Screen 14: First part of the practice episode.

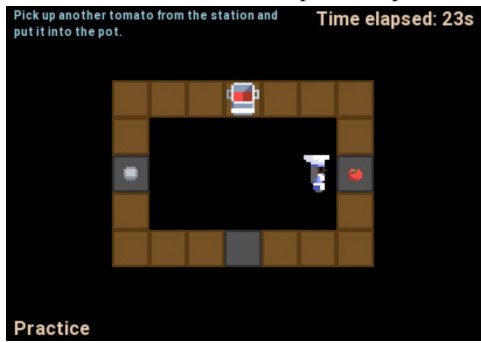

(c) Screen 14: Second part of the practice episode, after the participant has placed one tomato into the cooking pot.

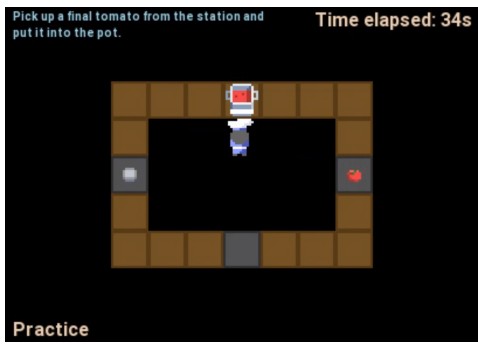

(d) Screen 14: Third part of the practice episode, after the participant has placed two tomatoes into the cooking pot.

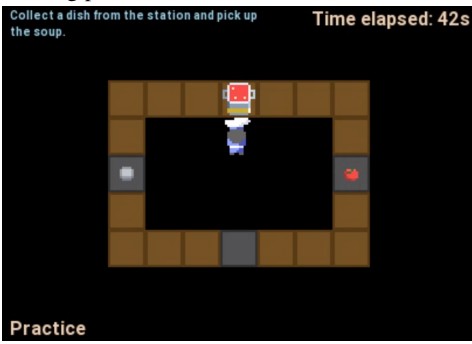

(e) Screen 14: Fourth part of the practice episode, after the participant has placed three tomatoes into the cooking pot.

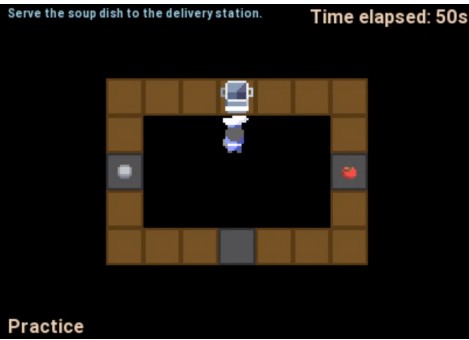

(f) Screen 14: Final part of the practice episode, after the participant has collected the soup from the cooking pot.

Figure 15: Screenshots of the practice episode.

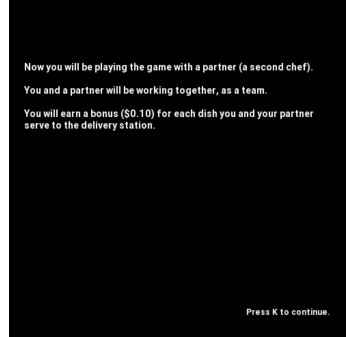

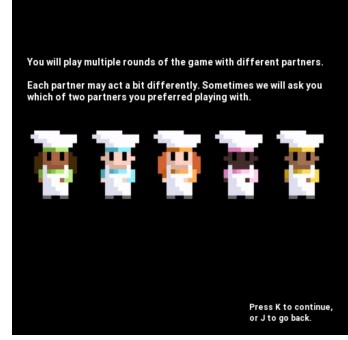

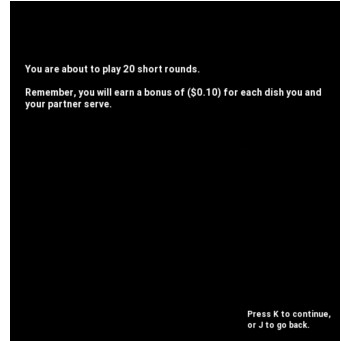

(a) Screen 15: Explain the payment structure for the experiment.

(b) Screen 16: Introduce the participants' partners.

(c) Screen 17: Provide overview of upcoming episodes.

Figure 16: Screenshots of instruction pages.

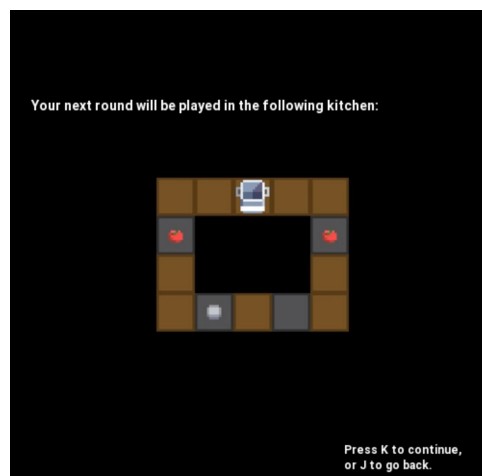

Screen 18: Introduce kitchen layout for first episode.

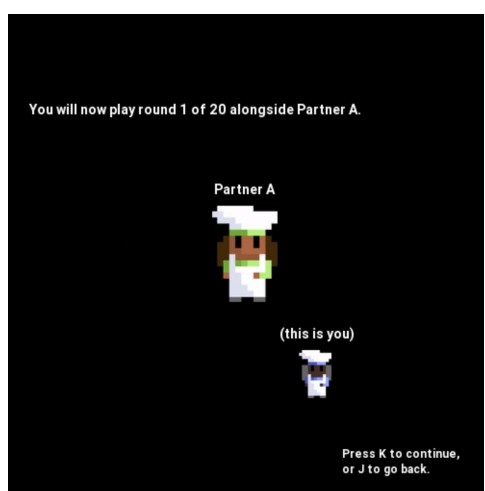

Screen 19: Introduce partner for first episode.

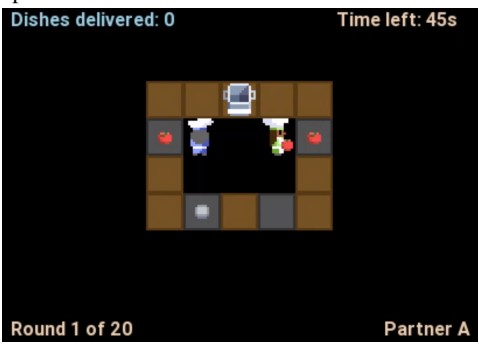

Screen 20: First episode.

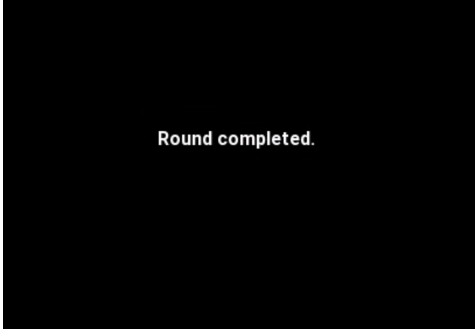

Screen 21: Confirm completion of first episode.

Figure 17: Screenshots of first episode.

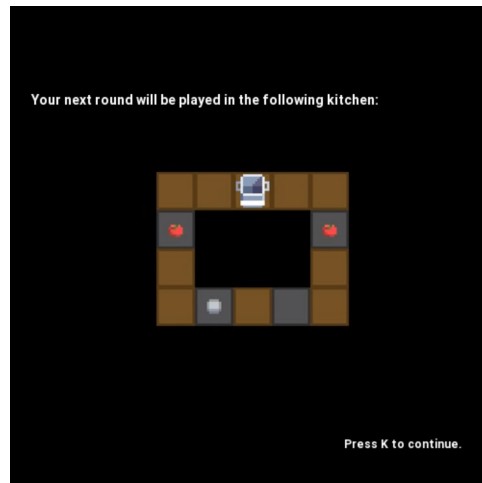

(a) Screen 22: Introduce kitchen layout for second episode.

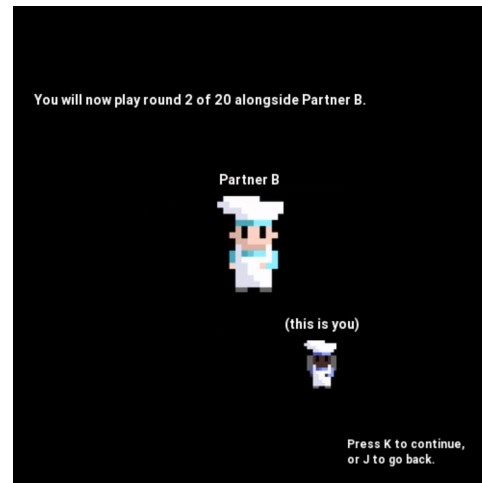

(b) Screen 23: Introduce partner for second episode.

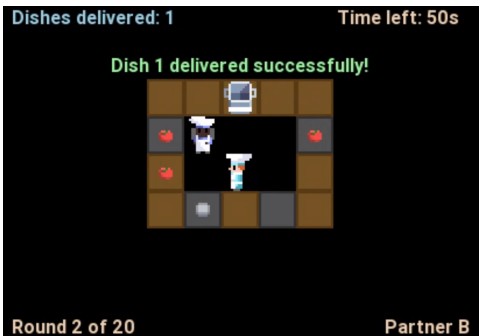

(c) Screen 24: Second episode.

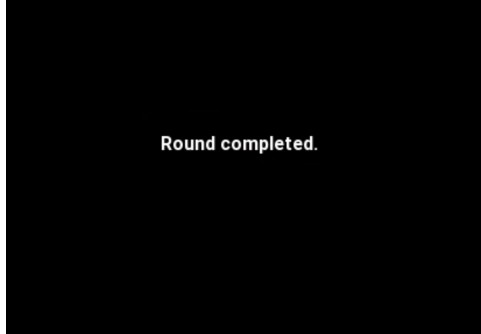

(d) Screen 25: Confirm completion of second episode.

Figure 18: Screenshots of second episode.

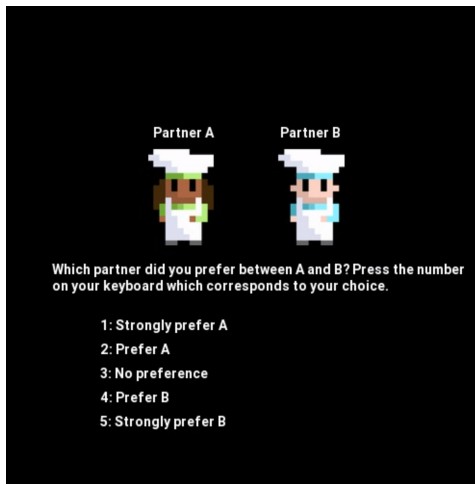

(a) Screen 26: Elicit participant's preference between partners.

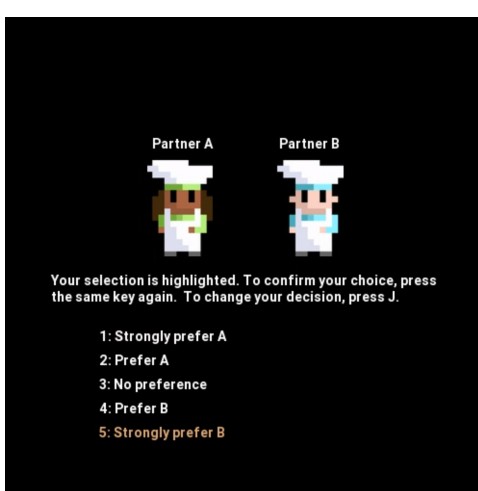

(b) Screen 27: Confirm participant's preference between partners.

Figure 19: Screenshots of preference elicitation over first and second episodes.

## D.2 Analytic details

We run a repeated-measures analysis of variance (ANOVA) to compare average team deliveries for each agent partner, with a random effect incorporated for each participant. Pairwise contrasts (using the Tukey method to adjust for multiple comparisons) indicated that the FCP agent achieved significantly higher delivery totals relative to the BCP agent, $t(1707) = 7.8$ ($p < 0.001$), the PP agent, $t(1707) = 8.2$ ($p < 0.001$), and the SP agent $t(1707) = 12.3$ ($p < 0.001$). The BCP agent also scored significantly higher than the SP agent, $t(1707) = 4.5$ ($p < 0.001$).

We conduct the same analysis to compare FCP against the $FCP_{-T}$ ablation. Human-agent teams involving the FCP agent completed significantly more deliveries than did those with the $FCP_{-T}$ agent, $t(797) = 14.4$ ($p < 0.001$).

To test whether participants preferred the FCP agent over other agent partners, we similarly fit a repeated-measures ANOVA on preferences elicited between FCP and any other agent, including the identity of the other agent as the sole predictor variable, as well as a random effect for each participant. Participants expressed significantly greater preferences for FCP over the SP agent, $t(329.6) = 4.9$ ($p < 0.001$), the PP agent, $t(399.0) = 2.5$ ($p = 0.011$), the BCP agent, $t(343.1) = 2.0$ ($p = 0.047$), and the $FCP_{-T}$ agent, $t(384.7) = 3.0$ ($p = 0.003$).

Finally, we run a repeated-measures ANOVA (again with a random effect for participants) to understand participants' preferences between the BCP agent and the PP agent. Participants reported significantly favoring the BCP agent over the PP agent, $t(77.8) = 3.1$ ($p = 0.003$).

## D.3 Additional quantitative results

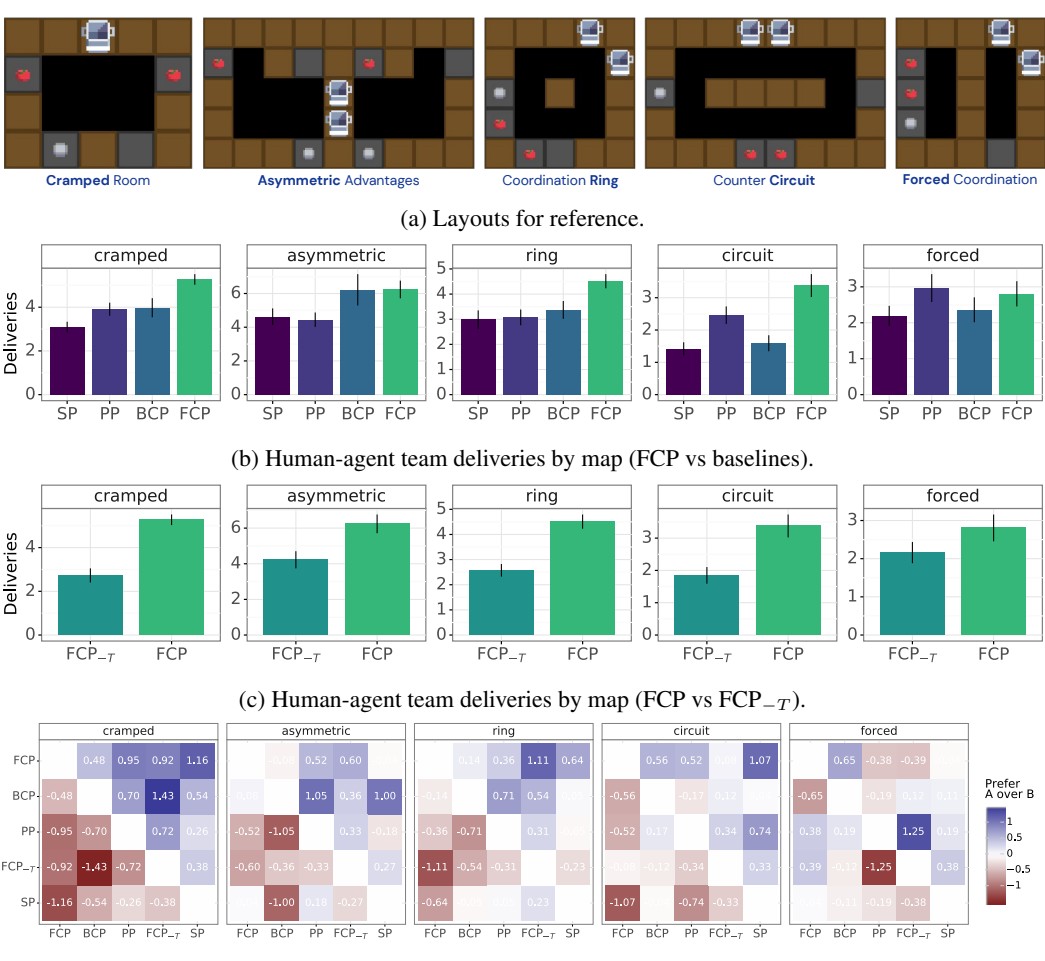

(a) Layouts for reference.

(b) Human-agent team deliveries by map (FCP vs baselines).

(c) Human-agent team deliveries by map (FCP vs FCP$_{-T}$).

(d) Human preferences for agent partners by map.

Figure 20: **Human-agent collaborative evaluation, per-layout:** Performance of each agent when partnered with human participants, as assessed by both objective and subjective metrics. In (a) and (b), error bars represent 95% confidence intervals. In terms of deliveries, FCP performs comparably or better than all baselines on every map. FCP is also the most consistently preferred agent across maps, though on some specific maps (asymmetric and forced) participants occasionally report preferring other agents over FCP.

## D.4 Additional qualitative results

At the end of the study, we prompted participants with an open-ended question for feedback on their partners. Many responses touched on adaptability, specialization, and goal compatibility. We include a sample of responses below.

- "Some did not adapt well to what I was doing or they were chaotic and I could not find a way to adapt to them."
- "Sometimes, depending on the layout of the kitchen, my partner seemed to chose a role and stick with it, allowing me to take the other role, and this seemed to be the most efficient way to work. For example, if my partner was 'in charge' of filling the pot, and I was in charge of putting it on a plate and delivering it. Also, I noticed that some of my partners seemed to know they needed to move around me, while others seemed to get 'stuck' until I moved out of their way."

- "I prefer partners which were willing not just to work independently but to let me hand them tomatoes/plates (since I maneuvered slower than them). It was difficult when they did not work on the same goal together but acted independently."

- "The responsive and compatibility of the other partners was really interesting as it took some quite some time to figure out what strategy works best for us and the strategy was almost instant with other partners."

- "Oh man I never laughed so much in my life. My partners were all fun to play with. Some were a bit faster than others. I think the one in pink was the fastest overall."

## E  Related work

Our work is similar to that conducted by Carroll et al. [12]. Here we provide a summary of the notable differences in experimental design that may contribute to differences in our results.

First, we did not use population-based training (PBT) with $N = 3$ as a baseline. We instead used population-play (PP) with $N = 32$, which shares some similarities with PBT. In our experiments, PP performed significantly better than SP. One potential reason for the performance difference between these two experiments is the small $N = 3$ population size used for PBT; this may have led to a collapse of diversity. This echos a key finding from Jaderberg et al.'s [74] methodological evaluation of PBT: "If the population size is too small (10 or below) we tend to encounter higher variance and can suffer from poorer results". This issue may have been exacerbated by the two-player common-payoff nature of our task.

Second, we trained our agents on an egocentric observation of the environment, as opposed to a top-down layer representation of the whole environment. Egocentric observations can improve agent generalization [75]. The fixed observation size in our approach additionally allowed us to train a single agent to play on all layouts, in contrast with Carroll et al.'s approach of training one agent per layout. We also used the VMPO learning algorithm [65] rather than PPO.

We implemented the game environment in the open-source environment engine DMLab2D [73]. Our implementation followed the implementation reported in the Carroll et al. [12], but was likely not a perfect recreation. We implemented our own human-agent interaction pipeline on top of the DMLab2D environment, with additional instructions and text to explain the game (as described in Section D). In our human-agent experiments, episodes lasted 300 steps over 60 seconds (5 FPS); in Carroll et al. [12], episodes lasted 400 steps over 60 seconds (6.66 FPS).

Finally, our BC agents were trained on 12,000 environment steps per layout, rather than 18,000 steps per layout, due to data collection limitations. We used an architecture with larger layers and an overall higher number of layers than used in Carroll et al. [12]. The observable features used to train the BC agent differed slightly; nonetheless, we observed no noticeable difference in performance. With our BC approach, we found that the random-action heuristic Carroll et al. [12] used was unnecessary.