# OpenReview forum: "Collaborating with Humans without Human Data"
_NeurIPS.cc/2021/Conference — NeurIPS 2021 Spotlight_

### Official Review · Reviewer_5uRi · 2021-07-14

**Rating:** 9
**Confidence:** 4

**Summary:**

This paper proposes using Fictitious Co-Play as a way to train agents so that they will be able to cooperate with humans. The idea is to avoid needing human data and rather to train a diverse set of self-play agents and then train an RL agent so it can play well with this diverse set of human surrogates. The paper is very well written, the method is simple and effective, and the results clearly illustrate the benefits of the proposed approach.

**Limitations And Societal Impact:**

Yes.

**Main Review:**

This is a very nice paper. I like the simplicity of the method and the thoroughness of the results and analysis. The authors do a good job of motivating the need for diversity and the fact that the method works well with humans despite not using any human data is seems very promising.  The experiments are well designed and the results are impressive.

Questions:
Line 125: Why was V-MPO chosen for RL? Would other methods work as well?
Line 137: Was this data collected from two different people playing for the first time or were they expert demos? Were the people cooperating well?

Figure 5a. It seems like BCP is the oracle here why does FCP perform so much better?





**Time Spent Reviewing:**

2

---

> ### Author Response · Authors · 2021-08-11
> **Initial response to Reviewer 5uRi**
>
> Thank you for your review, as well as your kind comments on the simplicity of our method and design of our study.
>
> To address your questions:
>
> > Line 125: Why was V-MPO chosen for RL? Would other methods work as well?
>
> We tested VMPO and A2C agents and found the former performed significantly better. To the best of our ability to compare, our VMPO agents seemed to also outperform the PPO agents in [1], suggesting they were performant. We anticipate that our method should be compatible with any performant RL algorithm, although we have not comprehensively tested this.
>
> > Line 137: Was this data collected from two different people playing for the first time or were they expert demos? Were the people cooperating well?
>
> We allowed participants limited practice first to ensure they understood the game, and qualitatively our impression was that they were playing well.
>
> > Figure 5a. It seems like BCP is the oracle here why does FCP perform so much better?
>
> While the BCP agent was trained with a BC agent, note that $H_\text{proxy}$ is another BC agent trained on a separate split of the human data, so this evaluation tests BCP’s generalization to what should be a similar, though not identical, partner. The conclusion is that BCP is quite brittle, a finding echoed in [2].
>
> [1] Carroll et al, On the utility of learning about humans for human-AI coordination, NeurIPS 2019 \
> [2] Knott et al, Evaluating the Robustness of Collaborative Agents, arXiv 2021

---

> > ### Comment · Reviewer_5uRi · 2021-09-01
> > **Re: Response**
> >
> > Thank you for your clarifications. I am maintaining my score of a strong accept.

---

### Official Review · Reviewer_bYPA · 2021-07-14

**Rating:** 5
**Confidence:** 4

**Summary:**

The paper introduces a practical method, named Fictitious Co-Play, for agents to learn to collaborate with humans / new agents in common-payoff games. The key idea is to first collect a diverse set of simulated agents, and then train the primary agent such that it can collaborate with the whole set of simulated partners. To promote the simulation diversity, they propose to gather the simulated agents at different learning stages, namely checkpoints, and from different runs, namely random seeds. The method is evaluated in the two-player cooking simulator with either held-out agents or human participants.

**Limitations And Societal Impact:**

Generally, I think the proposed method may inherit all the limitations of the domain randomization approach, e.g., the method may fail when the simulated sets of agents cannot cover the real-world human partner.

**Main Review:**

Overall:
The technical contribution seems limited. The experimental results are not fully convincing.

Originality:
The core idea of the paper seems very similar to domain randomization, an already popular technique for transferring policy from simulation to the real world. The task, collaborating with a human partner without human data, can be viewed as a particular instance of learning a policy for a real-world environment by using only simulation data. It's already known that diversifying the simulated domains can promote generalization. It would be good to see a clear discussion about how the proposed method differs from the previous domain randomization approach.

Quality:
The proposed method is highly intuitive. It would be good to see when it works and fails. Unfortunately, this is not clear in the experimental evaluations. In fact, half of the experiments are performed over held-out simulated agents, which are likely well represented by the simulated sets of agents during training.

Clarity:
The paper is well written and easy to follow.

Significance:
I'm not sure if the paper provides any significant insight into the community. Nevertheless, the simulation environment, which the author promised to open-source, might be a good contribution to the field.

[1] Tobin, Josh, et al. "Domain randomization for transferring deep neural networks from simulation to the real world." 2017 IEEE/RSJ international conference on intelligent robots and systems (IROS). IEEE, 2017.

------------------

Post rebuttal:

I've carefully read all responses from the authors and comments from other reviewers.

The authors addressed part of my concerns about experimental evaluations, and I, therefore, increase my rating from 4 to 5.

The limitation that prevents me from raising it to 6 or higher is that the technical contribution and insight seem quite marginal. After discussions with the authors and other reviewers, I still feel that the proposed method is a very natural instantiation of domain generalization in the considered human-AI collaboration context.

I suggest the authors add a detailed discussion about the connection between their method and the domain generalization framework, which in my opinion is essential.

**Time Spent Reviewing:**

11

---

> ### Author Response · Authors · 2021-08-11
> **Initial response to Reviewer bYPA**
>
> Thank you for your review, as well as your positive feedback on writing clarity.
>
> On domain randomization - great connection! We were indeed inspired in our approach by the success of domain randomization in robotics.  However, we disagree that the application of those ideas to zero-shot coordination is trivial, especially with humans. In typical domain randomization, varying the environment is as simple as sampling physical parameters from a range of values around their “true” estimated values. To apply this idea to randomizing collaborative partners, what parameters should be varied? The most direct analogy to domain randomization would be to build a BC model of humans and add noise to the fitted parameters. But this of course requires collecting human data, which we set out to avoid. What should be done instead then? Should we train RL agents to convergence, and then add parameter noise? Or should we instead train RL agents with a variety of hyperparameter choices? If the latter, which hyperparameters matter? The choices here are not obvious, and we think that reasoning through them and testing a subset in this domain constitutes quite a departure from domain randomization. Nevertheless, it is a great point that we should be more explicit about the relationship here. We originally drafted a discussion of the connection to this literature but removed it due to space constraints. We will try to fit it back in, at the very least in the Appendix.
>
> > It would be good to see when it works and fails. Unfortunately, this is not clear in the experimental evaluations. In fact, half of the experiments are performed over held-out simulated agents, which are likely well represented by the simulated sets of agents during training.
>
> First, as stated in the main text, our primary concern was the agent-human evaluations constituting the “other half” of the paper. The agent-agent results were intended as a cheap proxy to help guide our model choices and, practically speaking, are a necessary prerequisite to human-agent experiments. Comparable papers make the same choice (e.g. [1, 2, 3]). Second, our held-out simulated agents include both agents trained by BC and by RL, ensuring that those agents trained solely with BC partners (BCP) are tested with novel partners (RL agents) and those trained solely with RL partners (SP, PP, FCP) are tested with novel partners (BC agents). Third, evaluating solely with held-out simulated agents is a common element of zero-shot coordination studies (e.g. [4, 5]), so we believe our paper follows best practice in this field.
>
> > Nevertheless, the simulation environment, which the author promised to open-source, might be a good contribution to the field.
>
> We appreciate your enthusiasm on this point. Our environment has now been open-sourced, although sharing a link at this point would violate anonymity.
>
> [1] Carroll et al, On the utility of learning about humans for human-AI coordination, NeurIPS 2019 \
> [2] Hu et al, "Other-Play" for zero-shot coordination, ICML 2020 \
> [3] Tylkin et al, Learning Robust Helpful Behaviors in Two-Player Cooperative Atari Environments, NeurIPS Cooperative AI Workshop 2020 \
> [4] Knott et al, Evaluating the Robustness of Collaborative Agents, arXiv 2021 \
> [5] Lupu et al, Trajectory Diversity for Zero-Shot Coordination, ICML 2021

---

> > ### Comment · Reviewer_bYPA · 2021-08-12
> > **Questions & disagreements on the discussion of *domain randomization***
> >
> > Thanks for the response. I appreciate the discussion of the connection between the proposed method and domain randomization.
> >
> > > In typical domain randomization, varying the environment is as simple as sampling physical parameters from a range of values around their “true” estimated values.
> >
> > I'm afraid this statement is debatable.
> > * `domain randomization` has emerged as a whole field with thousands of variants. The key idea is to sample **simulated environments of diverse properties**, which are not necessarily physical parameters.
> > * In this sense, the considered setup is very much like a particular instance, where the `human co-player` is the `environment`, and the human `policy` is the key 'property'.
> >
> > > The most direct analogy to domain randomization would be to build a BC model of humans and add noise to the fitted parameters.
> >
> > * I cannot agree with the statement that `adding noise to the fitted parameters` is the most direct analogy to domain randomization. As noted above, the key idea of domain randomization is to perturb high-level properties instead of low-level parameters.
> > * Simply adding noise to model parameters does not really inherit the spirit of domain generalization, and will most likely fail due to the unrealistic and 'off manifold' simulation envs (policies).
> >
> > > To apply this idea to randomizing collaborative partners, what parameters should be varied? Or should we instead train RL agents with a variety of hyperparameter choices? If the latter, which hyperparameters matter? The choices here are not obvious, and we think that reasoning through them and testing a subset in this domain constitutes quite a departure from domain randomization.
> > * These are indeed open questions to explore. Nevertheless, I'm a bit skeptical about the claim `departure from domain randomization`. These questions are essentially about how to collect a diverse set of realistic envs (policies) from simulation. The proposed method (random seeds & random checkpoints) in my opinion is a fairly natural choice in the solution space.
> >
> > Overall, I agree with other reviewers that the paper presents an improved solution in the context of human-AI collaboration. Neverthelss, I still feel that (1) the considered problem is essentially a special case of sim-to-real transfer, and (2) the proposed solution is a natural instantiation of domain randomization.
> >
> > Will be glad to see different opinions from the authors and other reviewers about it.

---

> > > ### Author Response · Authors · 2021-08-13
> > > **Fictitious co-play as domain randomization**
> > >
> > > We thank the reviewer for their insightful comments and clear argument. We agree that our contribution can be viewed through the lens of domain randomization, and we will add citations to this literature in our introduction, arguing for diverse co-player policies as an important special case of diverse environments. This will help to build a stronger bridge between the typically distinct communities that focus on domain randomization and “fictitious play”-style algorithms.
> > >
> > > We believe that this connection serves to enhance the significance of our results, not reduce them. We have applied domain randomization in a very novel way to a problem that has not been addressed this way before, namely zero-shot generalization to human co-play in fully cooperative video games. We have demonstrated efficacy here which is surprising and significant with respect to prior work in the multi-agent literature (e.g. [1]), as identified by the other reviewers.
> > >
> > > If the reviewer knows of examples of domain randomization being applied to generate state-of-the-art ad-hoc collaboration between trained agents and unseen human partners in fully cooperative games, we’d be delighted to hear of this literature. To our knowledge, our method is unique in this respect.
> > >
> > > [1] Carroll et al, On the utility of learning about humans for human-AI coordination, NeurIPS 2019

---

> > > > ### Comment · Reviewer_bYPA · 2021-08-25
> > > > **Thanks for your response**
> > > >
> > > > Thanks for your response.
> > > >
> > > > I'm indeed not aware of any prior work applying the domain randomization framework to the human co-play context, as noted in my comments earlier.
> > > >
> > > > However, I'm still not fully convinced by ``applied domain randomization in a very novel way``, given that the proposed way is to use random seeds & random checkpoints.
> > > >
> > > > I will further discuss this with other Reviewers before finalizing my rating.

---

### Official Review · Reviewer_sJ2R · 2021-07-15

**Rating:** 8
**Confidence:** 4

**Summary:**

The paper studies the problem of cooperating with humans (without human data) in the overcooked domain, by proposing a new training procedure.  To cooperate with humans, they study making a training procedure for agents which robustly cooperate. The proposed procedure Fictitious Co-play (FCP) allows agents to robustly cooperate with other agents of varying skill-levels.  This is done by saving checkpoints of agents along training trajectories, and then training a final agent against a distribution of checkpoints.

**Ethical Concerns:**

I do not see ethical issues with the paper.

**Limitations And Societal Impact:**

The authors have adequately addressed the potential negative societal impacts of their work.

The authors present reasonable limitations.  A limitation that is not discussed, is that they introduce hyperparameters which are glossed over in the main paper.  Specifically, they must choose when to save checkpoints of the agents, and how to select which checkpoints the final agent will be trained against.  A (single) heuristic choice is made for Overcooked, but how well these choices will generalize to other games is unclear.

**Main Review:**

To the best of my knowledge, the proposed method is novel.  The differences from other competing methods -- ex., self-play, population-play, behavioral cloning -- is clear.  My experience is primarily in multi-agent learning and the related work looks adequately cited there.  I am less able to comment on related work for (a) zero-shot coordination, (b) reinforcement learning, or (c) human-agent interaction.

The paper does not provide a theoretical analysis of the method, though I am not familiar with any machinery the authors could use for analysis in zero-shot coordination, or coordination with humans.  A strength is the paper provides strong intuitive arguments for why the method works, which are supported by empirical results.  However, a weakness is the paper is highly focused on the Overcooked environment. Do these engineering choices generalize to other environments?  For example, do we ever need different choices about when to save checkpoints and how to select the checkpoints we train against, or can we simply use their 3 checkpoint heuristic?

For the most part, the paper is clearly written and the key ideas are easy to understand. The authors have excellent descriptions and visualizations for the overcooked environment. Some details in the appendix, like Algorithm 1 or a discussion of how checkpointing is handled, are important details and might be better suited to the main body. The tables are difficult to parse. For example, FCP, FCP_{-T, +A}, etc. are confusing notations for the idas they represent. Also, in figures (where space allows) I would appreciate it if you re-defined the acronyms that are used, so readers don’t have to look through the paper to find what they mean.  For example, H_proxy, notable baselines, or what the FCP subscripts intuitively represent. Alternatively, a notation table would be useful.

I believe the results are reasonably significant because zero-shot coordination, particularly with humans and limited data, will become increasingly important as machine learning models have to coordinate with people. Practitioners are likely to use the method, because it is simple to understand, does not look difficult to implement, does not introduce excessive computational burden, and seems to work reasonably well.  The paper claims to advance the state-of-the-art in coordinating with human partners in a set of gridworld tasks, which is supported by their human-agent experiments. The significance would be higher if the other domains besides Overcooked were evaluated.

My score is 8, because the authors provide a novel method with well-supported claims in a (for the most part) clearly written paper, which advances state of the art in the significant problem of coordinating with humans. My confidence 4, because I have experience with multi-agent learning, but less so with zero-shot coordination and human-agent interaction.
The score would be improved if authors investigated the method in domains besides overcooked, or clarity in the method, ablations and comparisons to baselines is improved.

**Time Spent Reviewing:**

10

---

> ### Author Response · Authors · 2021-08-11
> **Initial response to Reviewer sJ2R**
>
> Thank you for your review, including the positive comments on writing clarity, simplicity of the method, and empirical results.
>
> We agree that it will be interesting to see how FCP, and the particular hyperparameter settings used here, generalize to other domains. Intuitively, we expect that larger partner population sizes (N) will be needed for more complex games with larger spaces of performant strategies, and that more past checkpoints will be needed in games with more complex hierarchies of strategies to discover (e.g. hard exploration games with several rewarded subgoals). However, as you point out, this remains to be seen! We will better emphasize this limitation in the paper.
>
> > Some details in the appendix, like Algorithm 1 or a discussion of how checkpointing is handled, are important details and might be better suited to the main body.
>
> Thank you for this suggestion. We will do our best, given current space limitations. In particular, being more clear about checkpointing in the main text should be doable.
>
> > The tables are difficult to parse. For example, FCP, FCP_{-T, +A}, etc. are confusing notations for the ideas they represent.
>
> Agreed! However, we wanted short names in Table 1 that could later be reused in e.g. Figure 7b-7c. At the very least, we will try to remind the reader of their meaning in the caption, per your next suggestion. We will also see if we can fit more verbose subscripts, e.g. “FCP_{-T}” -> “FCP_{-past}” and “FCP_{+A}” -> “FCP_{+arch}”, or even natural language descriptions, e.g. “FCP_{-T}” -> “no past checkpoints” and “FCP_{+A}” -> “vary SP partner architectures.”
>
> > Also, in figures (where space allows) I would appreciate it if you re-defined the acronyms that are used, so readers don’t have to look through the paper to find what they mean. For example, H_proxy, notable baselines, or what the FCP subscripts intuitively represent. Alternatively, a notation table would be useful.
>
> Great idea! We had intended Figure 2 to serve as a sort of notation table, but we think it would be even better to remind the reader of each acronym in the captions. We will do our best to fit this in where possible, and add a notation table to the Appendix.

---

> > ### Author Response · Authors · 2021-09-01
> > **Following up with Reviewer sJ2R**
> >
> > Thank you again for thoughtful review and suggestions for improving our paper. As the end of the discussion period approaches, we wanted to check whether our response to your initial review was satisfactory and whether we can provide any additional clarifications that would change your initial score. We regret that the current conference format does not allow us to update the visible draft but assure you we will integrate your suggestions on notation into the final version.

---

> > > ### Comment · Reviewer_sJ2R · 2021-09-01
> > > **Inclined to maintain score**
> > >
> > > Thanks for your response with regards to my review.  I'm inclined to maintain my review of 8 with confidence 4 after thoroughly reading the other reviews and responses.  I am not currently seeking any clarifications for modifying the score, though I am happy about improving the notation because will help readers digest the work.

---

### Official Review · Reviewer_6sVb · 2021-07-21

**Rating:** 7
**Confidence:** 4

**Summary:**

The authors present a new training approach for human-AI collaboration (Fictitious Co-Play, FCP) which in a tested environment enables higher zero-shot coordination performance with humans than state-of-the-art. The idea behind FCP is to first train many self-play agents with different seeds and save snapshots from these training runs; then one can train the actual collaborative agent to perform well with all the previously saved snapshots. An important aspect of this approach is that it requires no human data, which can be expensive or difficult to come by. Experimentally, they show that this approach is able to perform both better than previous self-play or population-play approaches, and human aware agents (trained with a human model) trained with some amount of human data.

**Limitations And Societal Impact:**

While FCP clearly performs better than SP, PP, and BCP (at least in Overcooked), I am skeptical about the claim that the improvement is mostly due to improved coordination-protocol symmetry-breaking enabled by the FCP training.

Much of my skepticism originates in the fact that FCP is a suboptimal method even in the limit of perfect RL training (which I think should be emphasized more in the paper): to achieve best human-AI collaboration performance, one would optimally train an agent directly with humans (or with a perfect human model obtained with infinite data). Training with a diverse population might help increase the robustness of the trained agents, but will actually be biasing the method away from the true human-AI optimum. More on this below.

As mentioned in [Knott 2021], if one were to train with a population of adversarial agents (although we are in a collaborative setting), that would be far too conservative: the best thing in a self-driving domain might be to never leave the garage, because going outside you would almost inevitably crash (if everyone was trying to crash into you). Because of this, simply increasing diversity (the Diversity is All You Need approach, cited by this paper) will not necessarily increase performance (one instance of this might also be present in this paper itself, with the ablation introducing diversity of network architectures).

What is really needed for maintaining the theoretical claim to optimality is a population of _human-like_ agents. The more the population differs from human-likeness, the more the trained agents will be biased to act more conservatively or optimistically about their partner's capabilities than they possibly could.

Agents created by FCP will likely not be very human-like, both in Overcooked but especially so in environments like Hanabi. While you claim that "The ﬁnal checkpoint represents a fully-trained “skillful” partner, while earlier checkpoints represent less skilled partners.", Converged SP agents are not like skillful partners (the coordination protocols they settle on can be extremely uninterpretable and non-human-like), as you claim yourself in the intro and cite: [9, 10, 11, 18, 41, ...]. Earlier checkpoints too are likely not going to be similar to humans mid-way through learning about the game, as V-MPO agents and humans likely learn in very different ways: a beginner SP agent will be very different from a beginner human, and so doing well with the first will not guarantee doing well with humans. Additionally, humans definitely do not act randomly, but random agents (random initialization networks) are added to the population of FCP. This will definitely bias the obtained agents to be more independent and not rely on the other agent at first (under the assumption that the human might be random), more so than would be warranted by real humans.

Relatedly, as another source of bias, you are assuming that the pool of agents will be representative of all the conventions and symmetry breaking which humans perform too, and human biases (this is necessary for obtaining "a good agent partner [that] will adaptively switch between these conventions if a human clearly prefers one over the other"). Systematic myopia / cognitive biases would likely not be represented in the FCP population and could not be adapted to at test time. As an example, human-like ways of hinting in Hanabi would not necessarily naturally emerge among the SP trained agents. Generally, it might be very unlikely to stumble upon human-like protocols among the space of possible coordination protocols (this is most clear with languages, where training SP conversation bots with no grounding in languages would essentially never "stumble upon" english or any other human language – there is no way to get around this without human data / inductive biases).

However, as acknowledged above, clearly FCP is working better than just training with a SP agent or even an approximately learned human model (with limited human data). I believe that this is because actually FCP is improving the overall _robustness_ of the agents rather than simply their ability to adapt to different symmetry-breaking strategies (although this is probably improved too to some degree, as supported by Figure 8.b). I suspect that much of the improvement is due to decrease in brittleness relative to the baselines, which all overfit to the single agent they were trained with (the extreme brittleness is documented for Overcooked in [Knott 2021]). Many of the results are also consistent with the robustness hypothesis – for example also the fact that "FCP is able to move most frequently (35% of the time), corresponding to the best movement coordination with human partners". While this might be hard to verify (I can't think of an experiment in this setup off the top of my head), I would use the conversational AI example above as a cautionary tale: even in the limit of perfect RL convergence and compute, I doubt that doing FCP in a human-AI collaborative conversation task would work; English would never be "stumbled upon", and the agent – although very capable of adapting to optimal obscure languages that SP tends to come up with – would have no idea what to make of it. FCP in my mind might work empirically in many settings better than the alternatives, but is not guaranteed to and likely its performance will depend critically on the combination of domain and how the checkpointing is done (i.e. the population is formed).

Conditional on the authors being more clear about these limitations upfront and throughout the paper (i.e. 1. the inevitable bias that comes from not assuming access to human data, and 2. the difficulty of disambiguating robustness effects from increased coordination), I would be in favor of acceptance. This is a well executed work that sheds important insights on how one might build populations of agents that one can train with to improve human-AI performance.

**Main Review:**

## Originality

The method proposed is new, although it does have connections to previous work which also tries to construct populations of agents to improve collaboration performance [Knott, Evaluating the Robustness of Collaborative Agents, 2021] (not currently cited). It adapts similar ideas to those common in zero-sum settings for creating diverse populations to train high-performing agents, applying them in the collaborative setting.

## Quality / Clarity

The papers' claims are well supported, although in my opinion there is not enough emphasis on the limitations (as mentioned below). The experimental setup and the user study were well designed. Overall the paper was very well written and clear, and was a pleasure to read!

## Significance

These results constitute a significant contribution to the field of human-AI collaboration, which add to the conversation as to what are possible approaches to create AI agents able to collaborate with humans effectively. Through their results, the authors demonstrate that even without human data, in some domains we might be able to achieve better collaboration with humans than previous state of the art methods (which might even be using human data). Although their method is quite simple and not very dissimilar from previous population approaches in zero-sum settings, the results are strong enough to still make this a very worthwhile contribution that adds to the conversation.

**Time Spent Reviewing:**

4

---

> ### Author Response · Authors · 2021-08-11
> **Initial response to Reviewer 6sVb**
>
> Thank you for your detailed review, including your kind comments on writing clarity and experiment design, and your thorough suggestions for improving our discussion of limitations.
>
> For starters, thank you for the pointer to the very relevant work of Knott et al 2021 - the lack of citation / discussion was an oversight on our part. We will update our draft to cite them for: 1) also demonstrating the increased robustness that can come from training with more partners, 2) introducing variants of BCP that may potentially perform even better with humans, and 3) introducing a test suite that may provide a more detailed understanding of in what ways FCP is, and is not, robust.
>
> On limitations, we agree with your two high-level points.
>
> First, if the goal is to perform well with humans, then there is no perfect substitute for human data. FCP is a heuristic that happens to perform well for Overcooked, but to what extent our results and hyperparameter settings extend to other domains remains to be seen. For example, our intuitions about the application of FCP to emergent language are the same as yours - FCP might produce an agent capable of rapidly adapting to the kinds of random languages discovered by self-play (SP), but that agent would certainly have no special knowledge of or inclination towards English. The space of languages is too large for SP to stumble upon English, and nothing in the problem statement / reward function favors English. More generally, we certainly expect FCP to fail (and for human data to be essential) in settings where the reward function isn’t well-aligned with human play. In such settings (and in the absence of infinite human data), it may be interesting to explore hybrid approaches between FCP and BCP, that for example initialize FCP partners with BC (similar to what was done in AlphaGo and AlphaStar), or that simultaneously optimize both objectives. In any case, we will update our draft to be more clear about this point.
>
> Second, while our method was motivated by trying to make agents that are robust to symmetry-breaking conventions and skill levels, we agree that more work would be needed to 1) claim that our agents achieve these aspects of robustness specifically, and 2) claim that the random seeds and past checkpoints used for FCP partners approximate human variation in symmetry-breaking and skill level, respectively. On 1, unit tests a la Knott et al could be instructive, but as you suggest, designing the right one is not obvious. On 2, indeed we agree that certainly VMPO agents will not learn the game in the same way as humans, and so the argument for why this helps is likely more subtle. We will be sure to clarify this in our draft.
>
> In addition to improving the present work, we appreciate your detailed feedback in guiding how we think about future work in this domain as well.

---

> > ### Author Response · Authors · 2021-09-01
> > **Following up with Reviewer 6sVb**
> >
> > Thank you again for careful review and helpful comments. As the end of the discussion period approaches, we wanted to check whether our response to your initial review was satisfactory and whether we can provide any additional clarifications that would change your initial score. We regret that the current conference format does not allow us to update the visible draft but assure you we will integrate your feedback on limitations into the final version.

---

> > > ### Comment · Reviewer_6sVb · 2021-09-02
> > > **Maintaining score**
> > >
> > > Thanks for your clarifications and acknowledgement of the limitations! I am maintaining my score of accept.

---

### Decision · Program_Chairs · 2021-09-28

**Decision:**

Accept (Spotlight)

**Comment:**

This paper presents a simple trick to get multi-agent RL agents to cooperate effectively with a previously unseen agent (such as a human). The idea is to first train multiple agents with self-play and then train the final agent to cooperate with all of those agents as well as their past checkpoints. Experiments show successful results in the Overcooked game.

This submission was a pleasure to read. Three of the reviewers enthusiastically recommend acceptance, while the remaining reviewer recommends rejection because they consider it an incremental extension of domain randomization methods. In my opinion, this setting is substantially different from traditional uses of domain randomization, and while this work is certainly inspired by the idea of randomization helping generalization, the details are (as far as I know) novel. Apart from this, the reviewers didn't have any major objections. Since the submission seems high quality and ought to be of broad interest, I recommend it for a spotlight.


**Consistency Experiment:**

NeurIPS has a long history of experimentation. In 2014, NeurIPS ran an experiment in which 10% of submissions were reviewed by two independent committees to quantify the randomness in the review process. This year, we repeated a variant of this experiment to see how the quality of the review process has changed over time.  This paper was part of the experiment and was therefore assigned to two committees (consisting of reviewers, an Area Chair, and a Senior Area Chair) that reached independent decisions.  If both committees made the same recommendation, this recommendation was followed. If a single committee recommended acceptance, the paper was accepted (with the exception of a few cases in which the other committee identified what we considered a fatal flaw, e.g., an error in a key result).

This copy’s committee reached the following decision: **Accept (Spotlight)**

The other committee assigned to the paper recommended **Reject**.  You can find the other set of reviews, along with any follow up discussion with the authors here:
https://openreview.net/forum?id=79zWncwO2p